# A RADEMACHER-LIKE RANDOM EMBEDDING WITH LINEAR COMPLEXITY

## ABSTRACT

Random embedding assumes an important role in representation learning. Gaussian embedding and Rademacher embedding are two widely used random embeddings. Although they usually enjoy robustness and effectiveness, their computational complexity is high, i.e. $O(nk)$ for embedding an $n$-dimensional vector into $k$-dimensional space. The alternatives include partial subsampled randomized Hadamard (P-SRHT) embedding and sparse sign embedding, which are still not of linear complexity or cannot run efficiently in practical implementation. In this paper, a fast and robust Rademacher-like embedding (RLE) is proposed, based on a smaller Rademacher matrix and several auxiliary random arrays. Specifically, it embeds an $n$-dimensional vector into $k$-dimensional space in just $O(n)$ time and space (assuming $k$ is not larger than $O(n^{\frac{1}{2}})$). Our theoretic analysis reveals that the proposed RLE owns most of desirable properties of the Rademacher embedding while preserving lower complexity. To validate the practical efficiency and effectiveness, the proposed RLE is applied to single-pass randomized singular value decomposition (single-pass RSVD) for streaming data, and the randomized Arnoldi process based on sketched ordinary least-squares. Numerical experiments show that, with the proposed RLE the single-pass RSVD achieves 1.7x speed-up on average while keeping same or better accuracy, and the randomized Arnodli process enables a randomized GMRES algorithm running 1.3x faster on average for solving $Ax = b$ than that based on other embeddings.

## 1 INTRODUCTION

Random embedding, which projects high-dimensional vectors into low-dimensional vectors while still maintaining key properties, serves as a core tool for many fundamental representation learning and other related fields Woodruff (2014); Martinsson & Tropp (2020). Therefore, efficient and effective random embedding has attracted numerous research attention in recent years Tropp et al. (2019); Liberty (2013); Li et al. (2023). Notice that a synonym of random embedding is sketching, and random embedding has the same meaning as sketching in this work.

For handling high-dimensional data, random embedding is a required technique. Take singular value decomposition (SVD) as an example. When the dimension of data reaches millions or even billions, time cost and memory cost of classical SVD algorithm are intolerable. To tackle this problem, randomized SVD (RSVD) utilizes random embedding to project the high-dimensional data into much lower-dimensional space and then performs the decomposition Martinsson & Tropp (2020); Tropp et al. (2019); Yu et al. (2018); Feng et al. (2024); Yu et al. (2017)Musco & Musco (2015). Another example is Arnoldi process, which can be leveraged for obtaining eigenvalues/eigenvectors or solving linear equation systems. For large-scale data, the randomized Arnoldi process based on embedding is more preferable for both efficiency and robustness Balabanov & Grigori (2022).

The efficiency of random embedding plays an indispensable role. For example, random embedding in the single-pass RSVD Tropp et al. (2019) takes up majority of computing time. Besides, slow random embedding in the randomized Arnoldi process Balabanov & Grigori (2022) can lead to even worse performance than classical Arnoldi process.

Gaussian embedding is probably the most commonly-used embedding algorithm Gourru et al. (2022); He et al. (2015); Ren et al. (2016); Yüksel et al. (2021). It enjoys very good robustness,

and can be regarded as the most natural and simplest embedding algorithm Martinsson & Tropp (2020). In RSVD algorithms, Gaussian embedding is often adopted. Rademacher embedding is also a commonly-used embedding algorithm Verbin & Zhang (2012); Rakhshan & Rabusseau (2022). It can achieve similar embedding performance to Gaussian embedding, although its theoretic bounds may be a little bit worse than Gaussian embedding Martinsson & Tropp (2020) in certain scenarios. Gaussian embedding and Rademacher embedding are both dense embedding, which enjoys good robustness and little risk of embedding distortion. Their main shortcoming is that the complexity of projecting an $n$-dimensional vector into a $k$-dimensional vector is $O(nk)$.

To make embedding more efficient, fast transform techniques and several sparse embedding algorithms have been proposed Tropp et al. (2019); Martinsson & Tropp (2020). Partial subsampled randomized Hadamard (P-SRHT) embedding adopts fast Hadamard transform to make acceleration. For an $n \times k$ P-SRHT embedding matrix, time complexity of applying embedding one time can be $O(n \log n)$ or $O(n \log(2k))$. Sparse sign embedding is another alternative option. Notice that Count Sketch is a special kind of sparse sign embedding, with the highest sparsity. For small $k$, Count Sketch can lead to disastrous performance in practice Martinsson & Tropp (2020); Tropp et al. (2019), and so does the very sparse random projection method Li et al. (2006). An improved version of sparse sign embedding for practical applications was proposed in Tropp et al. (2019). For an $n \times k$ sparse sign matrix, time complexity of applying embedding one time can be $O(n)$. However, the trade-off of robustness and efficiency may restrict the application of sparse sign embedding. Moreover, it relies on sparse data structure and arithmetic which may affect actual efficiency due to irregular cache visiting Martinsson & Tropp (2020).

The aim of this work is to resolve the challenge of performing robust random embedding in lower complexity and practical efficiency. The main contribution of this paper can be concluded as follows.

(1) A Rademacher-like random embedding (RLE) approach is proposed, which utilizes a smaller Rademacher matrix and several random arrays to implicitly generate an embedding matrix and perform embedding. Particularly, it can embed an $n$-dimensional vector into $k$-dimensional space in just $O(n + k^2)$ time and space complexity, which is linear complexity if $k$ is not larger than $O(n^{\frac{1}{2}})$.

(2) Theoretical analysis of the proposed RLE approach is presented, where we prove its linear time and space complexity, that the square of the 2-norm of high-dimensional vector remains unchanged after embedding into low-dimensional space, and other properties as a random embedding.

(3) The proposed random embedding is employed in single-pass RSVD algorithm and randomized Arnoldi process. The latter facilitates a randomized GMRES algorithm. Numerical experiments show that with the proposed approach, the single-pass RSVD is accelerated by 1.7x on average while keeping same or even better accuracy, and the randomized GMRES is accelerated by 1.3x on average for the problems of solving $Ax = b$.

## 2 BACKGROUND

In this section, we will introduce the most commonly used random embedding methods and their typical applications to representation learning. For the readability of algorithms, we adopt MATLAB expression, e.g., $h_{1:j,1:j}$ corresponds to the submatrix consisting of the first $j$ rows and first $j$ columns. And throughout the paper, $\|\cdot\|$ denotes the 2-norm (spectral norm), while $\|\cdot\|_F$ denotes Frobenious norm.

### 2.1 RANDOM EMBEDDING

Random embedding aims to project a high-dimensional vector into low-dimensional space, i.e.

$$y = \Theta x, \tag{1}$$

where $x \in \mathbb{R}^n$, $\Theta \in \mathbb{R}^{k \times n}$ and $y \in \mathbb{R}^k$. Usually $k \ll n$ and one can assume $k$ is less than $O(n^{\frac{1}{2}})$. At the same time, $y$ should preserve key properties of $x$. It is required that the 2-norm of $y$ satisfy the following equation in mathematical expectation Martinsson & Tropp (2020):

$$\mathbb{E}[\|y\|^2] = \|x\|^2 . \tag{2}$$

If $\Theta_{i,j}$ obeys Normal$(0, k^{-1})$ distribution and each entry of $\Theta$ is independently identically distributed, then $\Theta$ is a Gaussian embedding matrix. It is noticeable that Gaussian embedding is regarded as the most natural embedding algorithms and have the most extensive applications Martinsson & Tropp (2020). it can be proven that the rescaled Gaussian embedding satisfies (2). The time and space complexity of Gaussian embedding is $O(nk)$.

If $\Theta_{i,j}$ is independently equal to $\frac{1}{\sqrt{k}}$ with probability 0.5 and is equal to $-\frac{1}{\sqrt{k}}$ otherwise, then $\Theta$ is a Rademacher embedding matrix. It can be proven that Rademacher embedding satisfies (2). Rademacher embedding can achieve similar performance as Gaussian embedding, making it favorable in many practical applications. However, the time complexity as well as space complexity of Rademacher sketching matrices is also $O(nk)$.

A robustness criterion of embedding relies on the concepts of $\epsilon$-subspace embedding and data oblivious embedding, which are explained in Appendix A.2. If the embedding matrix $\Theta$ is an $(\epsilon, \delta, d)$ oblivious $\ell_2 \to \ell_2$ subspace embedding, it can be applied to many basic algorithms with theoretic safety, e.g. the randomized Arnoldi process Balabanov & Grigori (2022). The Gaussian embedding and Rademacher embedding both satisfy this. The demand of matrix dimension for being an $(\epsilon, \delta, d)$ oblivious $\ell_2 \to \ell_2$ subspace embedding reflects the effectiveness of the embedding method. Theoretical results in Balabanov & Nouy (2019) shows that if $\Theta$ is a Rademacher matrix with $k \geq 7.87\epsilon^{-2}(6.9d + \log(1/\delta))$ for $\mathbb{R}$ and $k \geq 7.87\epsilon^{-2}(13.8d + \log(1/\delta))$ for $\mathbb{C}$, then $\Theta \in \mathbb{R}^{k \times n}$ is an $(\epsilon, \delta, d)$ oblivious $\ell_2 \to \ell_2$ subspace embedding.

To accelerate the process of random embedding, the sketching algorithms based on fast transform are developed. A famous example is P-SRHT embedding. If $\Theta$ denotes P-SRHT embedding matrix, $\Theta$ is the first $k$ row of SRHT embedding matrix $\Theta'$, where $\Theta'$ can be denoted with $\sqrt{\frac{n}{k}}DHS$, where $D$ is a diagonal matrix whose entries are 1 or -1, $H$ is a normalized Walsh-Hadamart matrix and $S$ is a randomly sampling matrix. P-SRHT embedding can be performed much faster, with time complexity of $O(n \log n)$ or $O(n \log k)$. However, P-SRHT embedding is less robust than Gaussian embedding or Rademacher embedding in certain scenarios.

Sparse sign embedding is even faster, with time complexity and space complexity of $O(n)$. If $\Theta$ can be denoted with $\sqrt{\frac{n}{C}}[s_1, \cdots, s_n]$ where $s_i$ are identically independently distributed random vectors with just $C$ non-zeros, then $\Theta$ is a sparse sign embedding matrix. Cohen (2016) pointed out that a sparse sign matrix is an oblivious subspace embedding with constant distortion for an arbitrary $d$-dimensional subspace of $\mathbb{R}^n$ when $C = O(\log d)$ and $k = O(d \log d)$. However, in many applications like RSVD, $d$ is large so that $C = O(\log d)$ and $k = O(d \log d)$ cause a time complexity higher than $O(n)$, and ultra sparse sketching (e.g. $C = 1$, equivalent to Count Sketch) may cause disastrous performance Tropp et al. (2019). To facilitate the practical applications of sparse sign embedding matrices, Tropp et al. (2019) proposed a practically applicable sparse sign embedding, which has the same form but $C = \min(k, 8)$ instead of $C = O(\log d)$. However, the above setting is heuristic, and in practice induces a non-negligible time constant. Moreover, another main disadvantage of sparse sign embedding is that we must use sparse data structures and arithmetic to achieve its benefits Martinsson & Tropp (2020).

Therefore, whether random embedding algorithm can be even faster (has linear time and space complexity with small time constant) and robust as well is still in question.

## 2.2 TYPICAL APPLICATIONS OF RANDOM EMBEDDING

In this subsection, we briefly introduce two important applications of random embedding algorithms: single-pass RSVD Tropp et al. (2019) and randomized Arnoldi process Balabanov & Grigori (2022). As the random embedding is the core of this paper, we only present the pseudo code and the use of random embedding in these algorithms.

The single-pass RSVD algorithm is shown in Algorithm 1, and in Tropp et al. (2019) the performance of the practical sparse sign embedding is tested. The single-pass RSVD algorithm is designed for processing streaming data, and also applicable when the data is too big to store in memory. The sketching process of the single-pass RSVD algorithm in Tropp et al. (2019) takes up the majority of time cost, where the acceleration of random embedding is of significance.

Random embedding also has application in the Krylov subspace iterative methods, e.g. the randomized Arnoldi process Balabanov & Grigori (2022) shown in Algorithm 2, which can be employed to

---

**Algorithm 1** Single-Pass Randomized SVD

---

**Input:** $A \in \mathbb{R}^{m \times n}$, rank parameter $k$, sketching parameter $r$, sketching parameter $s$.
**Output:** $U \in \mathbb{R}^{m \times k}$, $\Sigma \in \mathbb{R}^{k \times k}$, $V \in \mathbb{R}^{n \times k}$.

1: $\Gamma$, $\Delta$, $\Lambda$ and $\Xi$ are $r \times m$, $r \times n$, $s \times m$ and $s \times n$ random embedding matrices, respectively.
2: $X \leftarrow zeros(r, n)$, $Y \leftarrow zeros(m, r)$, $Z \leftarrow zeros(s, s)$.
3: **for** $i \leftarrow 1, 2, \cdots, m$ **do**
4:     $A_i \leftarrow$ the $i$-th part of $A$.          $\triangleright \sum_{i=1}^{m} A_i = A$ and $A_i$ only contains the $i$-th row of $A$.
5:     $X \leftarrow X + \Gamma A_i$, $Y \leftarrow Y + A_i \Delta^T$, $Z \leftarrow Z + \Lambda A_i \Xi^T$          $\triangleright$ random embedding.
6: **end for**
7: $Q \leftarrow qr(Y, 0)$          $\triangleright$ economic QR factorization.
8: $P \leftarrow qr(X^T, 0)$          $\triangleright$ economic QR factorization.
9: $C \leftarrow (\Lambda Q)^+ Z((\Xi P)^+)^T$
10: $[U, \Sigma, V] \leftarrow \mathbf{svd}(C, \text{'econ'})$          $\triangleright$ economic singular value decomposition.
11: $U \leftarrow QU(:, 1:k)$, $\Sigma \leftarrow \Sigma(1:k, 1:k)$, $V \leftarrow PV(:, 1:k)$

---

**Algorithm 2** The Randomized Arnoldi Process

---

**Input:** $A \in \mathbb{R}^{n \times n}$, $b \in \mathbb{R}^n$, $x_0 \in \mathbb{R}^n$, $m$, $\Theta \in \mathbb{R}^{k \times n}$.
**Output:** $V_{m+1} \in \mathbb{R}^{n \times (m+1)}$, $\overline{H}_m \in \mathbb{R}^{(m+1) \times m}$.

1: Calculate residual $r \leftarrow b - Ax_0$
2: Initialize $\overline{H}$ to be an $(m+1) \times m$ zero matrix
3: $s_1 \leftarrow \frac{\Theta r}{\|\Theta r\|}$,     $v_1 \leftarrow \frac{r}{\|\Theta r\|}$
4: **for** $j = 1, 2, \cdots, m$ **do**
5:     $w_j \leftarrow Av_j$
6:     $p_j \leftarrow \Theta w_j$          $\triangleright$ random embedding
7:     Solve $z_j = \arg\min_z \|S_j z - p_j\|$          $\triangleright$ solving sketched OLS problem
8:     $v_{j+1} \leftarrow w_j - V_j z_j$, with $V_j = [v_1, \cdots, v_j]$
9:     $s_{j+1} \leftarrow \Theta v_{j+1}$          $\triangleright$ random embedding
10:     $\overline{H}_{1:j+1,j} \leftarrow [z_j^T, \|s_{j+1}\|]^T$
11:     $v_{j+1} \leftarrow \frac{v_{j+1}}{\|s_{j+1}\|}$,     $s_{j+1} \leftarrow \frac{s_{j+1}}{\|s_{j+1}\|}$
12: **end for**
13: **Return** $V_{m+1} = [v_1, v_2, \cdots, v_{m+1}]$, $\overline{H}_m = \overline{H}$.

---

compute largest or smallest eigenvalues and eigenvectors, as well as to solve linear equation system. In the randomized Arnoldi process, sketched ordinary least-squares (OLS) problems are solved for orthogonalized basis vectors in Krylov subspace:

$$(\Theta V_j)z \approx \Theta w_j, \quad \text{i.e.} \quad \min_z \|(\Theta V_j)z - \Theta w_j\|, \tag{3}$$

where $\Theta \in \mathbb{R}^{k \times n}$ is a random embedding matrix and $k < n$. In Balabanov & Grigori (2022), P-SRHT is adopted to sketch, and Rademacher sketching is also recommended in Balabanov & Grigori (2022). The accuracy of the solution of the sketched OLS problem in Algorithm 2 directly affects the accuracy of Arnoldi process, and further the convergency of GMRES algorithm for solving $Ax = b$. Moreover, the sketching process takes up considerable time in randomized Arnoldi process and the GMRES iteration in the case of a sparse $A$ matrix.

In this paper, we will test our proposed random embedding on the above two applications to validate the practical efficiency of the proposed algorithm.

## 3 THE $O(n + k^2)$-COMPLEXITY RADEMACHER-LIKE EMBEDDING

In this section, we first introduce the idea of the proposed RLE approach. Then, the framework of RLE embedding, including setup phase and execution phase, is presented. After that, relevant theoretic analysis is presented.

## 3.1 THE IDEA

To obtain the dense random embedding algorithm with linear complexity, it is obvious that the embedding matrix can not be generated in explicit form, as the lowest time complexity for writing a $k \times n$ matrix is $O(nk)$. Thus we should generate the random embedding matrices implicitly.

For Rademacher embedding, the entries can only be $\frac{1}{\sqrt{k}}$ or $-\frac{1}{\sqrt{k}}$, which means that many entries in related positions have the same values. More particularly, any two rows of $\Theta \in \mathbb{R}^{k \times n}$ have $\frac{n}{2}$ same entries and $\frac{n}{2}$ different entries in mathematical expectation.

Given a $2 \times n$ Rademacher matrix, if we know which columns have same entries in the two rows beforehand, then we can calculate the sketching $y = \Theta x$ in just $n + 4$ flops instead of $2n$ flops (supposing the factor $\frac{1}{\sqrt{k}}$ multiplied afterward). The specific way is as follows. Firstly, we calculate the first partial sum of $x$ corresponding to the $\Theta$'s columns with same entries and then we calculate the second partial sum of $x$ on the other entries. While doing the partial sums, the signs of entries in the first row of $\Theta$ are considered. After that, we can directly add the first partial sum to the two rows of $y$, which was initialized to zero. Then, we can just add the second partial sum to the first row of $y$ and subtract the second partial sum from the other row of $y$. But how to know the positions of the columns with same entries beforehand? The answer is, we first assign these positions randomly.

The above simple case of $2 \times n$ matrix implies that we can utilize partial sum to fast calculate the result of Rademacher embedding. Therefore, aiming at accelerating the Rademacher embedding, we pursue a method which does not explicitly generate the embedding matrix, assigns the positions of the matrix columns with same entries in advance, and calculates the result of embedding leveraging the partial sums of $x$ multiplying $\frac{1}{\sqrt{k}}$ or $-\frac{1}{\sqrt{k}}$.

## 3.2 THE SETUP PHASE AND EXECUTION PHASE OF RLE

If the Rademacher matrix is in general $k \times n$, it is not as simple as the case of the $2 \times n$ matrix, because the positions on which two rows of the matrix have same entries depend on which two rows are considered. To overcome this difficulty, we propose a new random embedding method including two phases: the setup phase generates random vector, matrix and tensors, implicitly representing the embedding matrix; the execution phase calculates the result $y$ with the embedding.

The mechanism of the proposed embedding can be illustrated by Fig. 1, which explains how to calculate the $i$-th element of $y$ in (1). The idea starts from using a smaller Rademacher matrix $P \in \mathbb{R}^{\zeta \times n}$ and calculating random partial sums in $Px$ as building blocks for obtain the final result. Notice that $P$ has $\zeta$ rows, much fewer than $k$ rows, and each row of $P$ can be regarded as a basic random sequence. Then, for each row (element) of $Px$, the original sum of $n$ elements are randomly split to $k'$ partial sums. We set $k'$ to be a small multiple of $k$. These partial sums are stored in a temporary array $sum$ with $\zeta \times k'$ values. Notice the circle with blue border in Fig. 1 stands for a partial sum. This random splitting runs for $\xi$ times, so that $sum$ is actually a $\xi \times \zeta \times k'$ tensor. Again, $\xi$ is also a small number. Matrix $C \in \mathbb{Z}^{\xi \times n}$ is for realizing the random splitting, i.e., the $j$-th term in the sum for calculating $Px$ is accumulated into the $C(l, j)$-th column of $sum(l, :, :)$.

While calculating the $i$-th element of $y$, we first randomly choose which $\zeta \times k'$ partial sum array is to be used. Suppose this random choice is an index $R(i)$. The partial sums in $sum(R(i), :, :)$ are accumulated to get $y_i$. This accumulation is performed column by column, but for each column only one element with random row index is adopted. This random row index depends on which 2-D partial sum array is chosen, $i$'s value and which column is for accumulation. So, it is a $\xi \times k \times k'$ integer tensor, denoted by $E$. In Fig. 1, we assume for the specific $i$ the 2nd $\zeta \times k'$ partial sum array is chosen. It can be see that, the formation of partial sum array and the accumulation mechanism ensure that every element in $x$ has a contribution to $y_i$. Finally, while accumulating each partial sum, an extra random sign (1 or -1) is multiplied to enforce more randomness. The random signs are denoted by $S \in \mathbb{Z}^{\xi \times k \times k'}$, with same dimensions as $E$.

The pseudo codes for the execution phase of the proposed RLE approach (i.e. computing $y = \Theta x$) are shown in Algorithm 3. $\mathbb{Z}$ denotes the set of all integers.

From the above explanation and Alg. 3, we see that the proposed algorithm relies on the small-sized Rademacher matrix $P$, the random integer vector/matrix/tensor $R$, $C$ and $E$, and the random sign

Figure 1: Illustration of the proposed Rademacher-like embedding (RLE) algorithm ($\zeta = 2, \xi = 2$). $P$ is a $\zeta \times n$ Rademacher matrix. $R$, $C$ and $E$ are random integer vector, matrix and tensor, respectively. $S$ is a random sign tensor. Right: generation of partial sums; Left: calculation of $y_i$. Circle stands for an item in summation or a partial sum, and curved arrow depicts contribution to summation.

---

**Algorithm 3** Execution Phase of the Proposed Rademacher-Like Embedding

---

**Input:** $x \in \mathbb{R}^n$, Rademacher matrix $P \in \mathbb{R}^{\zeta \times n}$, random integer vector $R \in \mathbb{Z}^k$, random integer matrix $C \in \mathbb{Z}^{\xi \times n}$, random integer tensor $E \in \mathbb{Z}^{\xi \times k \times k'}$, random sign tensor $S \in \mathbb{Z}^{\xi \times k \times k'}$.
        $\triangleright$ $R$'s value range is $[1, \xi]$, $C$'s value range is $[1, k']$, $E$'s value range is $[1, \zeta]$
**Output:** $y \in \mathbb{R}^k$.                  $\triangleright$ $y = \Theta x$.

1: Initialize $y$ to the zero vector, and $sum$ to the zero tensor $\in \mathbb{R}^{\xi \times \zeta \times k'}$     $\triangleright$ $k' = \omega k$
2: **for** $i = 1, 2, \cdots, \xi$ **do**
3:   **for** $j = 1, 2, \cdots, \zeta$ **do**
4:     **for** $l = 1, 2, \cdots, n$ **do**
5:       $u \leftarrow C_{i,l}$
6:       $sum_{i,j,u} \leftarrow sum_{i,j,u} + x_l \times P_{j,l}$        $\triangleright$ calculate the partial sums.
7:     **end for**
8:   **end for**
9: **end for**
10: **for** $i = 1, 2, \cdots, k$ **do**
11:   **for** $j = 1, 2, \cdots, k'$ **do**
12:     $a \leftarrow R_i, \quad b \leftarrow E_{a,i,j}, \quad c \leftarrow S_{a,i,j}$
13:     $y_i \leftarrow y_i + sum_{a,b,j} \times c$     $\triangleright$ accumulate the partial sums with random signs.
14:   **end for**
15: **end for**
16: **Return** $y = [y_1, y_2, \cdots, y_k]^T$.

---

tensor $S$. So, generating them constitutes the setup phase of RLE approach. For all the random vector/matrix/tensor used, we let them follow the uniform distribution in their value range.

A key point of the proposed method is that the relevant dimension parameters are all small numbers. $\zeta$ and $\xi$ are usually integers no more than 3. And, $k' = \omega k$, where $\omega$ is also usually no more than 3. This guarantees the low complexity of performing this random embedding. Finally, we should pointed out the proposed method is equivalent to multiplying a random matrix $\Theta$ and $x$, where $\Theta$ is very similar to Rademacher matrix. We will explain this in next subsection, after proving some good properties of this embedding matrix $\Theta$.

## 3.3 THEORETIC ANALYSIS

Firstly, we analyze the computational complexity of RLE.

**Theorem 1.** *The time complexity and space complexity of proposed RLE approach are both $O(n + k^2)$. If $k$ is not larger than $O(n^{\frac{1}{2}})$, they are linear complexity.*

*Proof.* We divide the proof into two parts: complexity for setup and complexity for execution.

The setup phase includes the generation of a $\zeta \times n$ Rademarcher matrix $P$, a $k$-dimensional random integer vector $R$, a $\xi \times n$ random integer matrix $C$, and two $\xi \times k \times \omega k$ random tensors ($E$ and $S$). Because $\zeta$, $\xi$ and $\omega$ are small constants, the time complexity and space complexity of the setup phase of RLE approach are all $O(n + k^2)$.

For execution phase, from Alg. 3 we see that the calculation of $sum$ is of $O(\xi\zeta n)$ time complexity and $O(\xi\omega k)$ space complexity. The calculation of $y$ costs $O(\omega k^2)$ time and $O(k)$ space. So, the time complexity of the execution phase is $O(n + k^2)$ and the space complexity is $O(k)$.

To summarize, the time complexity and space complexity of proposed RLE approach are both $O(n + k^2)$. If $k$ is not larger than $O(n^{\frac{1}{2}})$, $O(n + k^2)$ becomes $O(n)$, which is a linear complexity. □

The properties of the embedding matrix $\Theta$ implicitly generated by the proposed RLE are as follows.

**Theorem 2.** *Suppose $\Theta \in \mathbb{R}^{k \times n}$ is the random matrix implicitly generated with the proposed RLE approach. Then, every entry of $\Theta$ is $\frac{1}{\sqrt{k}}$ or $-\frac{1}{\sqrt{k}}$.*

*Proof.* From Alg. 3 or Fig. 1 we can see, with the proposed RLE approach $y_i$ is a linear combination of all elements of $x$, and the combinational coefficients are the elements in $P$ multiplied by a random sign. As $y_i$ equals to the $i$-th element of $\Theta x$ and $P$ is a Rademacher matrix with elements being $\frac{1}{\sqrt{k}}$ or $-\frac{1}{\sqrt{k}}$, the combinational coefficients, i.e. entries in $\Theta$, must be $\frac{1}{\sqrt{k}}$ or $-\frac{1}{\sqrt{k}}$. □

**Theorem 3.** *Suppose $\Theta \in \mathbb{R}^{k \times n}$ is the random matrix implicitly generated with the proposed RLE approach, $\Theta_{i,j}$ and $\Theta_{l,r}$ are any two different entries in $\Theta$. Then, $\Theta_{i,j}$ and $\Theta_{l,r}$ are independent to each other in mathematics. In other words, the entries in $\Theta$ are pairwisely independent. Moreover, the rows of $\Theta$: $r_1, r_2, \cdots, r_k$ are mutually independent, and the entries in the same row of $\Theta$ are mutually independent.*

*Proof.* From Alg. 3 or Fig. 1 we can see that, the entry of $\Theta$ is the entry of Rademarcher matrix $P$ multiplied by a random sign (from $S$). If $j \neq r$, $\Theta_{i,j}$ and $\Theta_{l,r}$ correspond to two different entries in $P$ (in column $j$ and $r$ respectively). Thus, $\Theta_{i,j}$ and $\Theta_{l,r}$ are independent of each other. Otherwise, $i \neq l$ must hold. In this case, the signs (from $S$) for $\Theta_{i,j}$ and $\Theta_{l,r}$ are independently multiplied. Thus, $\Theta_{i,j}$ and $\Theta_{l,r}$ are also independent to each other. Also due to the independence of the signs (from $S$), $r_1, r_2, \cdots, r_k$ are mutually independent. Moreover, the entries in the same row of $\Theta$ are mutually independent, because they depend on different entries of $P$. □

**Theorem 4.** *Each entry in $\Theta \in \mathbb{R}^{k \times n}$ which is generated implicitly with the proposed RLE approach has a probability of 0.5 being $\frac{1}{\sqrt{k}}$ and being $-\frac{1}{\sqrt{k}}$. Moreover, $\mathbb{E}[\Theta^T\Theta] = I$, where $I$ denotes the identity matrix, and the diagonal element of $\Theta^T\Theta$ is always 1.*

*Proof.* From Alg. 3 we see that, each entry of $\Theta$ is lastly multiplied by a random sign from $S$. Let $p_{i,j}^*$ denote the probability of $\Theta_{i,j}$ being $\frac{1}{\sqrt{k}}$ before this last operation. Let $p_{i,j}$ denote the probability of $\Theta_{i,j}$ being $\frac{1}{\sqrt{k}}$ after the last operation. Then we have (also based on Theorem 2):

$$p_{i,j} = 0.5 \times p_{i,j}^* + 0.5 \times (1 - p_{i,j}^*) = 0.5. \tag{4}$$

Therefore, each entry in $\Theta \in \mathbb{R}^{k \times n}$ which is implicitly generated by the above algorithm has 0.5 probability of being $\frac{1}{\sqrt{k}}$ and being $-\frac{1}{\sqrt{k}}$. Let $F$ denote $\Theta^T\Theta$. Then, its diagonal elements satisfy

$$F_{i,i} = \sum_{o=1}^{k} \Theta_{o,i}\Theta_{o,i} = \sum_{o=1}^{k} \Theta_{o,i}^2 = \sum_{o=1}^{k} \frac{1}{k} = 1. \tag{5}$$

For non-diagonal elements of $F$, we have:

$$\mathbb{E}[F_{i,j}] = \mathbb{E}[\sum_{o=1}^{k} \Theta_{o,i}\Theta_{o,j}] = \sum_{o=1}^{k} \mathbb{E}[\Theta_{o,i}\Theta_{o,j}] = 0, \tag{6}$$

where the last equality is due to each entry of $\Theta$ has 0.5 probability of being $\frac{1}{\sqrt{k}}$ and being $-\frac{1}{\sqrt{k}}$ and the pairwise independence stated in Theorem 3. Therefore, we have $\mathbb{E}[\Theta^T\Theta] = I$. $\square$

The proofs of the following theorems are given in Appendix A.1 and A.2, respectively.

**Theorem 5.** *Suppose $\Theta \in \mathbb{R}^{k \times n}$ is a random matrix generated implicitly with the proposed RLE approach and $x$ is an $n$-dimensional vector. Then, $\mathbb{E}[\|\Theta x\|^2] = \|x\|^2$.*

**Theorem 6.** *Suppose $0 < \epsilon < 0.572$, $0 < \delta < 1$ and $\Theta \in \mathbb{R}^{k \times n}$ is the matrix generated implicitly with the proposed RLE approach. If $k \geq 7.87\epsilon^{-2}(6.9d + \log(\frac{1}{\epsilon}))$ for $\mathbb{R}$ and $k \geq 7.87\epsilon^{-2}(13.8d + \log(\frac{1}{\epsilon}))$ for $\mathbb{C}$, $\Theta$ is an $(\epsilon, \delta, d)$ oblivious $\ell_2 \to \ell_2$ subspace embedding.*

Theorem 6 states the proposed RLE ensures theoretic safety, and its statement about $(\epsilon, \delta, d)$ oblivious $\ell_2 \to \ell_2$ subspace embedding is the same as that for the Rademacher embedding Balabanov & Nouy (2019). A difference between the proposed RLE and the Rademacher embedding is that the mutual independence of whole matrix entries does not hold for the former. This is the reason that we name the proposed approach a Rademacher-like embedding. It is noticed that, the proposed RLE algorithm has an $O(n + k^2)$ computational complexity, which is a linear complexity when $k$ is not larger than $O(n^{\frac{1}{2}})$. This is a remarkable advantage for practical usage.

## 4 NUMERICAL EXPERIMENTS

The proposed RLE approach is applied to the single-pass randomized SVD and randomized Arnoldi process to evaluate its effectiveness and efficiency. The parameter $\xi$, $\zeta$, $\omega$ are set to 2, 1, 2, respectively. All programs are implemented in C++ and Intel MKL, and the experiments are conducted on a single core of a computer with Xeon Gold 6230R CPU@2.10GHz and 128 GB RAM.

### 4.1 RESULTS ON SINGLE-PASS RSVD WITH STREAMING DATA

We have implemented the single-pass RSVD algorithm in Tropp et al. (2019). And, the random embedding steps there are implemented with Gaussian embedding, sparse sign embedding, and the proposed RLE embedding, respectively. The test matrices include those generated synthetically following Martinsson & Tropp (2020) and those from Feng et al. (2023). Their dimensions are listed in Table 1.

Particularly, "noise" is generated by:

$$\text{diag}(1, \cdots, 1, 0, \cdots, 0) + 10^{-4}GG^*, \tag{7}$$

where the number of 1 is 20 and $G$ is a standard Gaussian matrix. "pol" and "exp" are random matrices whose singular values satisfy the following two formulas respectively:

$$\sigma_i = \begin{cases} 1 & i \leq 20 \\ (i-19)^{-2} & i > 20 \end{cases} \tag{8}$$

$$\sigma_i = \begin{cases} 1 & i \leq 20 \\ 10^{-0.1(n-20)} & i > 20 \end{cases} \tag{9}$$

"Weather" are directly generated by the open-source code of Tropp et al. (2019). "inv" and "sqrt" are two random matrices whose singular values satisfy $\sigma_i = \frac{1}{i}$ and $\sigma_i = \frac{1}{\sqrt{i}}$, respectively. "MNIST" is from Lecun et al. (1998), and "FeretMat" is a dense matrix generated based on Phillips et al. (2000).

The parameter $k$ is set to 10 just like that in the numerical experiments in Tropp et al. (2019). Moreover, the parameter $r$ is set to $k + 1$ and the parameter $s$ is set to $2r + 1$ as is suggested by Tropp et al. (2019). Particularly, $r = 11$ and $s = 23$ in our experiments. Below are the two metrics of the relative error of RSVD.

$$err_s = \frac{\|A - A_k\| - \|A - A_k^*\|}{\|A - A_k^*\|} \tag{10}$$

Table 1: Results of the single-pass RSVD (Alg. 1) with different random embedding approaches.

| Case | $m$ | $n$ | Gaussian Embedding | | | Sparse Sign Embedding | | | Proposed RLE | | | | |
|------|-----|-----|--------------------|--|--|------------------------|--|--|--------------|--|--|--|--|
| | | | $T_{tot}$(s) | $err_s$ | $err_f$ | $T_{tot}$(s) | $err_s$ | $err_f$ | $T_{tot}$(s) | $err_s$ | $err_f$ | $Sp_1$ | $Sp_2$ |
| noise | 1.0E3 | 1.0E3 | 0.0863 | 31.3 | 6.53 | 0.112 | 2.70 | 0.37 | 0.0850 | 6.92 | 0.78 | 1.02 | 1.32 |
| exp | 1.0E3 | 1.0E3 | 0.0978 | 3.89 | 1.02 | 0.111 | 2.80 | 0.83 | 0.0800 | 2.31 | 0.70 | 1.22 | 1.38 |
| pol | 1.0E3 | 1.0E3 | 0.0863 | 1.35 | 0.58 | 0.113 | 1.83 | 0.71 | 0.0750 | 1.81 | 0.66 | 1.15 | 1.50 |
| Weather | 1.9E4 | 7.3E3 | 3.04 | 17.0 | 0.31 | 3.60 | 43.7 | 0.74 | 1.78 | 27.7 | 0.50 | 1.71 | 2.02 |
| MNIST | 6.0E4 | 7.8E2 | 1.10 | 3.91 | 1.01 | 1.47 | 2.94 | 0.76 | 0.900 | 3.42 | 0.79 | 1.22 | 1.63 |
| inv | 4.0E4 | 4.0E4 | 32.0 | 2.69 | 1.41 | 33.8 | 6.45 | 2.10 | 18.1 | 5.58 | 1.90 | 1.77 | 1.87 |
| sqrt | 4.0E4 | 4.0E4 | 31.9 | 7.66 | 0.72 | 34.0 | 5.83 | 0.53 | 19.0 | 8.70 | 0.69 | 1.68 | 1.79 |
| FERET | 1.0E5 | 3.9E5 | 1248 | - | - | 1179 | - | - | 617 | - | - | 2.02 | 1.91 |
| Average | | | | 9.69 | 1.65 | | 9.46 | 0.86 | | 8.06 | 0.86 | 1.5 | 1.7 |

$m$ and $n$ are the dimensions of test matrix, $T_{tot}$ denote the total time for performing single-pass randomized SVD (time for reading data is omitted), $err_s$ and $err_s$ denote the relative error in 2-norm and Frobenious norm, respectively. $Sp_1$ and $Sp_2$ denote the speed-up ratios of the proposed RLE based single-pass RSVD algorithm over those based on Gaussian sketching and sparse sign sketching, respectively.

$$err_f = \frac{\|A - A_k\|_{\mathrm{F}} - \|A - A_k^*\|_{\mathrm{F}}}{\|A - A_k^*\|_{\mathrm{F}}} \quad (11)$$

Here $A_k^*$ denotes the best rank $k$ approximation matrix to $A$ and $A_k$ denotes the rank $k$ approximation to $A$ generated by the RSVD algorithm. As the computation of $A_k^*$ for "FERET" causes memory overflow in our machine, we could not obtain the relative errors for "FERET".

The results are listed as Table 1. From it, we can see the single-pass RSVD equipped with the proposed RLE shows 1.5x average speed-up and 1.7x average speed-up compared with those equipped with the Gaussian random embedding and sparse sign embedding, respectively. It is noticeable that in this experiment, Gaussian embedding is coded by matrix-vector multiplication function in MKL, which makes it even faster than sparse sign embedding in small test cases. Even though, the proposed RLE approach shows prominent advantages. Moreover, proposed algorithm also shows good accuracy, whose relative 2-norm error is remarkably smaller than Gaussian embedding and sparse sign embedding.

### 4.2 RESULTS ON RANDOMIZED ARNOLDI PROCESS

We have implemented a randomized GMRES algorithm which combines the standard GMRES Saad & Schultz (1986) and the randomized Arnoldi process (Alg.2) Balabanov & Grigori (2022). The embedding process utilizes P-SRHT embedding, sparse sign embedding and the proposed RLE. The P-SRHT embedding utilizes the efficient open-source code in SRH. The standard GMRES Saad & Schultz (1986) is also implemented for comparison. They are used to solve linear equation $Ax = b$ and the test cases are from SparseSuite Matrix Collection mat. Particularly, "rajat31" is a case with a 4.7E6×4.7E6 sparse coefficient matrix. The matrix dimensions of Case "memchip" and "circuit5M" are 2.7E6×2.7E6 and 5.6E6×5.6E6, respectively. And, more experimental results and the ablation study are presented in Appendix A.3 and A.4. We use the decrease trends of relative residual $\frac{\|Ax-b\|}{\|b\|}$ to evaluate the performance of different versions of GMRES, which are shown in Figure 2. The embedding dimension $k$ is set to 200, and ILU(3) factorization is applied as the preconditioning .

From the experimental results, we can see that with the proposed approach, randomized Arnoldi process can proceed much faster than the standard Arnoldi process and P-SRHT based randomized Arnoldi process. As the standard GMRES does not have a setup phase, its time cost is low in first a few of iteration steps. However, the high-dimensional Gram-Schmidt orthogonalization makes the standard GMRES run slowly in the latter iteration steps. As the sparse sign embedding has comparably larger time constant and may lose data locality, it does not show prominent advantage over the efficiently implemented P-SRHT. Compared with P-SRHT and sparse sign, the proposed RLE approach shows prominent advantages. On the test cases, the randomized GMRES with proposed RLE approach shows 1.4x, 1.3x and 1.3x speed-ups on average over the standard Arnoldi process, the P-SRHT based randomized GMRES and the sparse sign embedding based randomized GMRES respectively, for reaching convergence or the maximum of 100 iterations. Moreover, the

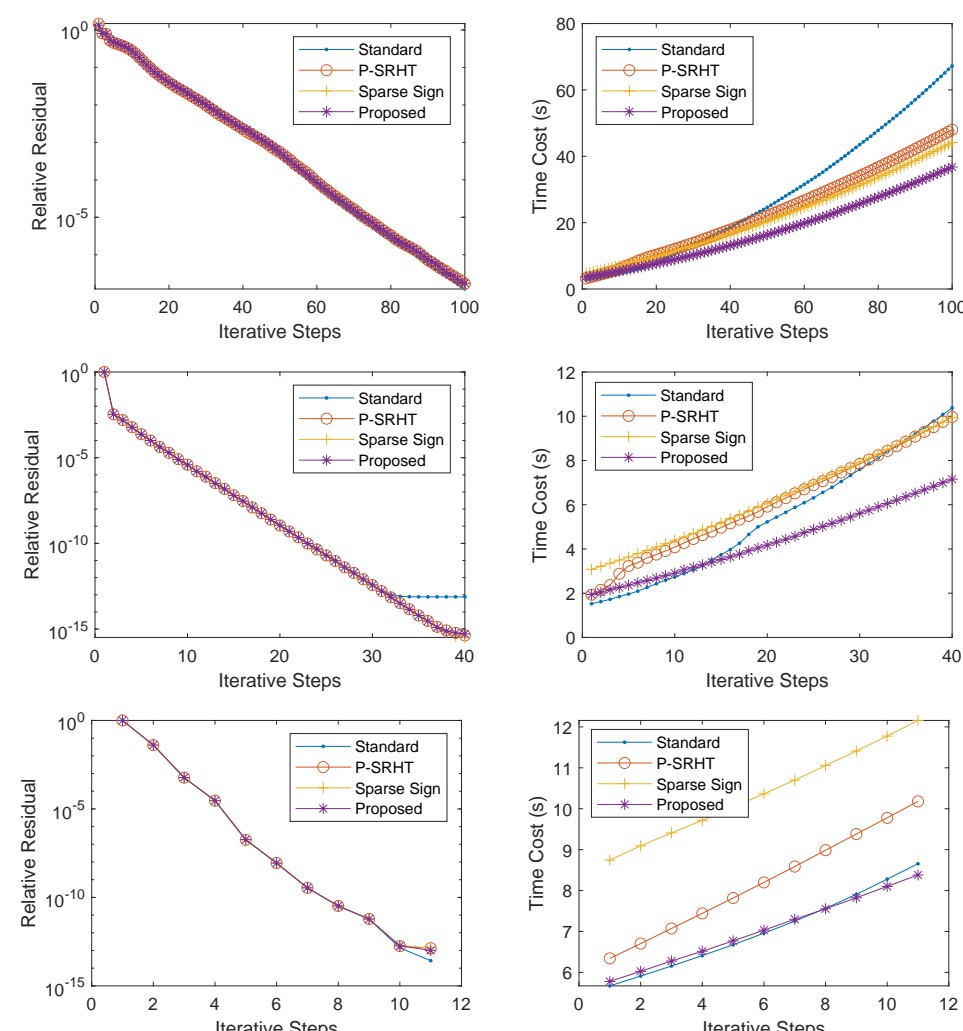

Figure 2: Convergence behaviors and runtime trends of the original GMRES and the randomized GMRES algorithms utilizing the randomized Arnoldi process (Alg. 2) with different embedding approaches for "rajat31", "memchip", "circuit5M" (from top to bottom respectively).

convergence rate of the randomized Arnoldi process with the proposed RLE is very similar to that of standard Arnoldi process, which indicates the robustness of proposed RLE.

## 5 CONCLUSION

In this paper, a fast and robust random embedding algorithm, named Rademacher-like embedding (RLE), is proposed. The theoretic analysis of the proposed approach is given, which proves it is a Rademacher-like random embedding with $O(n + k^2)$ computational complexity for embedding an $n$-dimensional vector into $k$-dimensional space. The numerical experiments on single-pass RSVD show that, compared with the sparse sign embedding the proposed RLE enables 1.7x average speed-up with same or even better accuracy. And the experiments on randomized Arnoldi process shows that the proposed RLE enables an accelerated GMRES algorithm which runs 1.3x faster on average without the loss of accuracy.

In the future, we will explore the parallel version of RLE approach. From Fig. 1 we can see the calculation of $y$ can be easily parallelized. For the generation of partial sums, some skills can also apply to make it well parallelized.

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

## A  APPENDIX

### A.1  PROOF OF THEOREM 5

*Proof.* Firstly, we have

$$\mathbb{E}[\|\Theta x\|^2] = \mathbb{E}[x^T \Theta^T \Theta x]. \tag{12}$$

As $\Theta$ is independently generated without $x$, we have

$$\mathbb{E}[\|\Theta x\|^2] = \mathbb{E}[x^T]\mathbb{E}[\Theta^T \Theta]\mathbb{E}[x]. \tag{13}$$

With Theorem 4, we have that $\mathbb{E}[\Theta^T \Theta] = I$. Therefore, we can derive

$$\mathbb{E}[\|\Theta x\|^2] = \mathbb{E}[x^T]\mathbb{E}[x]. \tag{14}$$

As $x$ is deterministic, we further derive

$$\mathbb{E}[\|\Theta x\|^2] = x^T x = \|x\|^2, \tag{15}$$

which completes the proof. $\square$

### A.2  DEFINITIONS ON OBLIVIOUS SUBSPACE EMBEDDING AND PROOF OF THEOREM 6

First of all, we introduce the definitions on oblivious subspace embedding (OSE). Then we will prove Theorem 6 with the help of Theorem 3, and the proof inherits from the proof of the (Balabanov & Nouy, 2019, Proposition 3.7).

#### A.2.1  THEORETICAL ASPECTS OF RANDOM EMBEDDING

In theory, the embedding matrix $\Theta$ is desired to have some theoretic properties. For numerical stability of random embedding, the distortion after embedding is often estimated in probability. With the following two definitions, embedding distortion can be depicted in theory.

**Definition 1.** *Given $\epsilon < 1$, the embedding matrix $\Theta \in \mathbb{R}^{k \times n}$ is an $\epsilon$-subspace embedding for subspace $\mathbb{K} \subset \mathbb{R}^n$, if for any vectors $x, y \in K$,*

$$|\langle x, y \rangle - \langle \Theta x, \Theta y \rangle| \le \epsilon \|x\| \cdot \|y\|, \tag{16}$$

*where $\langle x, y \rangle$ denotes the inner product of vector $x$ and $y$.*

**Definition 2.** *The embedding matrix $\Theta \in \mathbb{R}^{k \times n}$ is an $(\epsilon, \delta, d)$ oblivious $\ell_2 \to \ell_2$ subspace embedding, if it is an $\epsilon$-subspace embedding for any fixed $d$-dimensional subspace $\mathbb{K} \subset \mathbb{R}^n$ with probability at least $1 - \delta$.*

If $\Theta$ is an $(\epsilon, \delta, d)$ oblivious $\ell_2 \to \ell_2$ subspace embedding, $\Theta$ can be applied to many basic algorithms with theoretic safety. For example, theoretic results in Balabanov & Grigori (2022) shows that if $\Theta$ is an $(\epsilon, \delta, d)$ oblivious $\ell_2 \to \ell_2$ subspace embedding, numerical stability of randomized Arnoldi process can be guaranteed.

Meanwhile, to extend the embedding for the field of real numbers to the field of complex numbers, a very useful proposition was proposed in the supplementary material of Balabanov & Nouy (2019).

**Proposition 1.** *(Balabanov & Nouy, 2019, supplementary material) Let $\Theta$ be a real random matrix. If $\Theta$ is $(\epsilon, \delta, 2d)$ oblivious $\ell_2 \to \ell_2$ subspace embedding for subspaces of vectors in $\mathbb{R}^n$, then it is $(\epsilon, \delta, d)$ oblivious $\ell_2 \to \ell_2$ subspace embedding for subspaces of vectors in $\mathbb{C}^n$.*

The dimension of embedding matrices for $(\epsilon, \delta, d)$ oblivious $\ell_2 \to \ell_2$ subspace embedding is often used to evaluate the effectiveness of different embedding methods. Theoretical results in Balabanov & Nouy (2019) shows that if $\Theta$ is a rescaled Rademacher matrix with $k \ge 7.87\epsilon^{-2}(6.9d + \log(1/\delta))$ for $\mathbb{R}$ and $k \ge 7.87\epsilon^{-2}(13.8d + \log(1/\delta))$ for $\mathbb{C}$, then $\Theta \in \mathbb{R}^{k \times n}$ is an $(\epsilon, \delta, d)$ oblivious $\ell_2 \to \ell_2$ subspace embedding. Theorem 6 states that the same error bound holds for the proposed RLE approach.

### A.2.2 THE PROOF OF THEOREM 6

In this subsection, we use $\Theta$ to denote the embedding matrix generated by the RLE approach and $r_i$ to denote the $i$-th row of $\Theta$. Define $Q_i(\alpha)$ for an arbitrary $n$-dimensional vector $\alpha$ as:

$$Q_i(\alpha) = \frac{\sqrt{k}}{\sqrt{n}} r_i \alpha \tag{17}$$

With Theorem 3, we can see that $Q_i(\alpha)$ is mutually independent. So,

$$\mathbb{E}[Q_i(\alpha)] = \mathbb{E}[\sum_{j=1}^{n} \alpha_j \sqrt{\frac{k}{n}} r_{i,j}] = \sum_{j=1}^{n} \mathbb{E}[\alpha_j \sqrt{\frac{k}{n}} r_{i,j}] = \sqrt{\frac{k}{n}} \sum_{j=1}^{n} \alpha_j \mathbb{E}[r_{i,j}] = 0 , \tag{18}$$

$$\mathbb{E}[Q_i(\alpha)^2] = \mathbb{E}[(\sum_{j=1}^{n} \alpha_j \sqrt{\frac{k}{n}} r_{i,j})^2] = \frac{k}{n} \mathbb{E}[\sum_{j=1}^{n} (\alpha_j r_{i,j})^2 + \sum_{l=1}^{n} \sum_{j=1, j \neq l}^{n} (\alpha_l \alpha_j r_{i,l} r_{i,j})]$$

$$= \frac{k}{n} (\mathbb{E}[\sum_{j=1}^{n} (\alpha_j r_{i,j})^2] + \mathbb{E}[\sum_{l=1}^{n} \sum_{j=1, j \neq l}^{n} (\alpha_l \alpha_j r_{i,l} r_{i,j})]) = \frac{k}{n} (\sum_{j=1}^{n} \frac{1}{k} \alpha_j^2 + 0) \tag{19}$$

$$= \frac{1}{n} \|\alpha\|^2 .$$

Define $W(\alpha)$ as:

$$W(\alpha) = \sum_{i=1}^{k} (\sqrt{\frac{k}{n}} r_i \alpha)^2 = \sum_{i=1}^{k} Q_i^2(\alpha) . \tag{20}$$

Below we first prove Lemma 5.2 and Lemma 5.1 in Achlioptas (2003) still hold for the RLE approach.

**Lemma 1.** *For all $h \in [0, \frac{n}{2})$, all $n \geq 1$ and all unit vectors $\alpha$,*

$$\mathbb{E}[\exp(hQ_1(\alpha)^2)] \leq \frac{1}{\sqrt{1 - \frac{2h}{n}}}, \ \ \mathbb{E}[\exp(Q_1(\alpha)^4)] \leq \frac{3}{n^2}. \tag{21}$$

*Proof.* Based on the proof of Lemma 5.2 of Achlioptas (2003), we see that (21) holds if the entry in $r_1$ is mutually independent and has 0.5 probability to be 1 and 0.5 probability to be -1. Because the $r_1$ derived from the RLE satifies these conditions (see Theorem 3), (21) holds. $\square$

**Lemma 2.** *For any $\epsilon > 0$ and any unit vector $\alpha \in \mathbb{R}^n$,*

$$\Pr[W(\alpha) > (1 + \epsilon) \frac{k}{n}] < \exp(-\frac{k}{2}(\frac{\epsilon^2}{2} - \frac{\epsilon^3}{3})),$$

$$\Pr[W(\alpha) < (1 - \epsilon) \frac{k}{n}] < \exp(-\frac{k}{2}(\frac{\epsilon^2}{2} - \frac{\epsilon^3}{3})). \tag{22}$$

*Proof.* The proof can be naturally inherited from that for Lemma 5.1 in Achlioptas (2003), because we can leverage the mutual independence of $Q_1(\alpha), Q_2(\alpha), \cdots, Q_k(\alpha)$ derived from the proposed RLE.

For arbitrary $h$, we can derive:

$$\Pr[W(\alpha) > (1 + \epsilon) \frac{k}{n}] = \Pr[\exp(hW(\alpha)) > \exp(h(1 + \epsilon) \frac{k}{n})] = \Pr[\frac{\exp(hW(\alpha))}{\exp(h(1 + \epsilon) \frac{k}{n})} > 1]$$

$$< \mathbb{E}[\exp(hW(\alpha) \exp(-h(1 + \epsilon) \frac{k}{n})] = \exp(-h(1 + \epsilon) \frac{k}{n}) \mathbb{E}[\exp(hW(\alpha)] \tag{23}$$

With the mutual independence of $Q_1(\alpha), Q_2(\alpha), \cdots, Q_k(\alpha)$, it is obvious to see $Q_1(\alpha), Q_2(\alpha), \cdots, Q_k(\alpha)$ are identically independently distributed (i.i.d.). Thus we have:

$$\mathbb{E}[\exp(hW(\alpha))] = \mathbb{E}[\prod_{i=1}^{k} \exp(hQ_i^2)] = \prod_{i=1}^{k} \mathbb{E}[\exp(hQ_i^2)] = (\mathbb{E}[\exp(hQ_1(\alpha)^2)])^k \tag{24}$$

Thus, we have:

$$\Pr[W(\alpha) > (1+\epsilon)\frac{k}{n}] < \exp(-h(1+\epsilon)\frac{k}{n})(\mathbb{E}[\exp(hQ_1(\alpha)^2)])^k \quad (25)$$

With Lemma 1 and given $h = \frac{n}{2}\frac{\epsilon}{1+\epsilon} < \frac{n}{2}$, we have:

$$\Pr[W(\alpha) > (1+\epsilon)\frac{k}{n}] < \exp(-h(1+\epsilon)\frac{k}{n})(\frac{1}{\sqrt{1 - \frac{2h}{n}}})^k = ((1+\epsilon)\exp(-\epsilon))^{\frac{k}{2}}$$
$$< \exp(-\frac{k}{2}(\frac{\epsilon^2}{2} - \frac{\epsilon^3}{3})) \quad (26)$$

The last inequality is due to $-\frac{k}{2}(\epsilon\log(1+\epsilon)) < -\frac{k}{2}(\frac{\epsilon^2}{2} - \frac{\epsilon^3}{3})$. Furthermore, we can similarly derive that:

$$\Pr[W(\alpha) < (1-\epsilon)\frac{k}{n}] < \exp(h(1-\epsilon)\frac{k}{n})(\mathbb{E}[\exp(-hQ_1(\alpha)^2)])^k \quad (27)$$

$$\Pr[W(\alpha) < (1-\epsilon)\frac{k}{n}] < \exp(h(1-\epsilon)\frac{k}{n})(\mathbb{E}[1 - hQ_1(\alpha)^2 + \frac{(-hQ_1(\alpha)^2)^2}{2!}])^k$$
$$= \exp(h(1-\epsilon)\frac{k}{n})(1 - \frac{h}{n} + \frac{h^2}{2}\mathbb{E}[Q_1(\alpha)^4])^k \quad (28)$$

With Lemma 1 and given given $h = \frac{n}{2}\frac{\epsilon}{1+\epsilon} < \frac{n}{2}$, we have:

$$\Pr[W(\alpha) < (1-\epsilon)\frac{k}{n}] \leq \exp(h(1-\epsilon)\frac{k}{n})(1 - \frac{h}{n} + \frac{3}{2}(\frac{h}{n})^2)^k$$
$$= (1 - \frac{\epsilon}{2(1+\epsilon)} + \frac{3\epsilon^2}{8(1+\epsilon)^2})^k \exp(\frac{\epsilon(1-\epsilon)k}{2(1+\epsilon)}) \quad (29)$$
$$< \exp(-\frac{k}{2}(\frac{\epsilon^2}{2} - \frac{\epsilon^3}{3}))$$

$\square$

Below we present the proof of Theorem 6. The idea of proof is naturally inherited from the proof of Proposition 3.7 in Balabanov & Nouy (2019).

*Proof.* This proof can be naturally inherited from the proof of Proposition 3.7 in Balabanov & Nouy (2019), with the help of the above Lemmas.

We first consider embedding for $\mathbb{R}$. The proof follows the standard framework in Woodruff (2014). Given a $d$-dimensional subspace $V \subseteq \mathbb{R}^n$, $\mathcal{O} = \{x \in V : \|x\| = 1\}$ be the unit sphere of $V$. According to the (Bourgain et al., 1989, Lemma 2.4), for any $\gamma > 0$ there exists a $\gamma-$net $\mathcal{N}$ of $\mathcal{O}$ satisfying $\#\mathcal{N} \leq (1 + \frac{2}{\gamma})^d$. For $0 < \eta < \frac{1}{2}$, let $\Theta$ be the RLE embedding matrix with $k \geq 6\eta^{-2}(2d\log(1 + \frac{2}{\gamma}) + \log(\frac{1}{\delta}))$. With Lemma 2 and an union bound argument, we obtain for a fixed $x \in V$

$$\Pr(|\|x\|^2 - \|\Theta x\|^2| \leq \eta\|x\|^2) \geq 1 - 2\exp(\frac{-k\eta^2}{6}). \quad (30)$$

Therefore, by leveraging an union bound for the probability of success, we have:

$$\{|\|x + y\|^2 - \|\Theta(x + y)\|^2| \leq \eta\|x + y\|^2, \forall x, y \in \mathcal{N}\} \quad (31)$$

holds with probability at least $1 - \delta$. We can further derive that:

$$\{|\langle x, y \rangle - \langle \Theta x, \Theta y \rangle| \leq \eta, \forall x, y \in \mathcal{N}\} \quad (32)$$

holds with probability at least $1 - \delta$. Let $a$ be some vector in $\mathcal{O}$. Assuming $\gamma < 1$, it can be proven by induction that $b = \sum_{i \geq 0} \alpha_i a_i$, where $a_i \in \mathcal{N}$ and $0 \leq \alpha_i \leq \gamma^i$. Then we can derive:

$$\|\Theta a\|^2 \leq \sum_{i,j \geq 0} \langle \Theta a_i \Theta a_j \rangle \alpha_i \alpha_j \leq \sum_{i,j \geq 0} (\langle a_i, a_j \rangle \alpha_i \alpha_j + \eta\alpha_i\alpha_j) = 1 + \eta(\sum_{i \geq 0} \alpha_i)^2 \leq 1 + \frac{\eta}{(1-\gamma)^2}$$
$$(33)$$

Similarly, we can obtain:

$$\|\Theta a\|^2 \geq 1 - \frac{\eta}{(1-\gamma)^2} \tag{34}$$

Therefore, if (38) holds, we have:

$$|1 - \|\Theta a\|^2| \leq \frac{\eta}{(1-\gamma)^2}. \tag{35}$$

For a given $\epsilon \leq \frac{1}{2(1-\gamma)^2}$, let $\eta = (1-\gamma)^2\epsilon$. Since (32) holds for an arbitrary vector $a \in \mathcal{O}$, using the parallelogram identity, we can derive:

$$|\langle x, y \rangle - \langle \Theta x, \Theta y \rangle| \leq \epsilon \|x\| \cdot \|y\| \tag{36}$$

holds for all $x, y \in V$ if (32) holds. Thus we conclude that $\Theta$ with $k \geq 6\epsilon^{-2}(1-\gamma)^{-4}(2d\log(1+\frac{2}{\gamma})+\log(\frac{1}{\delta}))$ is an $(\epsilon, \delta, d)$ obvious $\ell_2 \to \ell_2$ subspace embedding for $V$ with probability at least $1-\delta$. The lower bound of the number of rows of $\Theta$ is obtained by taking $\gamma = \arg\min_{x \in (0,1)}(\frac{\log(1+\frac{2}{x})}{(1-x)^4}) \approx 0.0656$. For $\mathbb{C}$, we can leverage the Proposition 1 for the proof. If $\Theta$ is an $(\epsilon, \delta, 2d)$ obvious embedding for $\mathbb{R}$, then $\Theta$ is an $(\epsilon, \delta, d)$ obvious embedding for $C$. Thus, we complete the proof of Theorem 6. □

## A.3 MORE EXPERIMENT RESULTS ON RANDOMIZED ARNOLDI PROGRESS AND GMRES

Experiments on all test cases of randomized GMRES is listed in Table 2. To validate the efficiency of our proposed RLE approach in more industrial scenarios, we test the proposed RLE approach on famous power grid benchmarks, ibmpg Nassif and thupg Yang & Li. The test cases include "ibmpg4t" which is a 1.2E6 × 1.2E6 sparse matrix with 4.8E6 non-zeros and "thupg5" which is a 2.0E7 × 2.0E7 sparse matrix with 8.8E7 non-zeros.

Table 2: Results of the randomized GMRES with different random embedding approaches.

| Case | $n$ | $nnz$ | $t_{GMRES}$ | $t_P$ | $t_S$ | $t_{RLE}$ | $Sp_1$ | $Sp_2$ | $Sp_3$ |
|---|---|---|---|---|---|---|---|---|---|
| rajat31 | 4.7E6 | 2.0E7 | 67.3 | 48.0 | 44.2 | 36.8 | 1.8 | 1.3 | 1.2 |
| memchip | 2.7E6 | 1.5E7 | 10.4 | 9.97 | 9.95 | 7.16 | 1.5 | 1.4 | 1.4 |
| circuit5M | 5.6E6 | 6.0E7 | 8.66 | 10.2 | 12.2 | 8.38 | 1.0 | 1.2 | 1.5 |
| ibmpg4t | 1.2E6 | 4.8E6 | 17.8 | 13.1 | 12.8 | 10.1 | 1.8 | 1.3 | 1.3 |
| thupg5 | 2.0E7 | 8.8E7 | 385 | 254 | 221 | 185 | 2.1 | 1.4 | 1.2 |
| Average | | | | | | | 1.6 | 1.3 | 1.3 |

$nnz$ is the number of non-zeros. $t_{GMRES}$ denotes the runtime of standard GMRES, $t_P$ denotes the runtime of randomized GMRES with P-SRHT embedding, $t_S$ denotes the runtime of randomized GMRES with sparse sign embedding, $t_{RLE}$ denotes the runtime of randomized GMRES with the proposed RLE. $Sp_1$, $Sp_2$, $Sp_3$ are the speed-ups brought by the proposed RLE algorithm to the standard GMRES, randomized GMRES with P-SRHT embedding, randomized GMRES with sparse sign embedding, respectively.

The experimental results are shown in Fig.3 and Fig.4, where we can see that the proposed RLE approach is prominently faster than P-SRHT embedding and sparse sign embedding. Particularly, on test case "ibmpg4t", randomized GMRES with the proposed RLE approach shows 1.8x speed-up, 1.3x speed-up, 1.3x speed-up compared with standard GMRES, randomized GMRES with P-SRHT embedding, randomized GMRES with sparse sign embedding, respectively. On test case "thupg5", randomized GMRES with the proposed RLE approach shows 2.1x speed-up, 1.4x speed-up, 1.2x speed-up compared with standard GMRES, randomized GMRES with P-SRHT embedding, randomized GMRES with sparse sign embedding, respectively. Meanwhile, we can see that the randomized GMRES with proposed RLE achieve similar convergence rate with standard GMRES, which indicates the robustness of the proposed RLE in industrial scenarios.

## A.4 ABLATION STUDY ON PARAMETERS FOR RLE

There are three parameters $\xi$, $\zeta$ and $\omega$ in RLE. We use the randomized Arnoldi process based on RLE as an example to do the ablation study on these parameters.

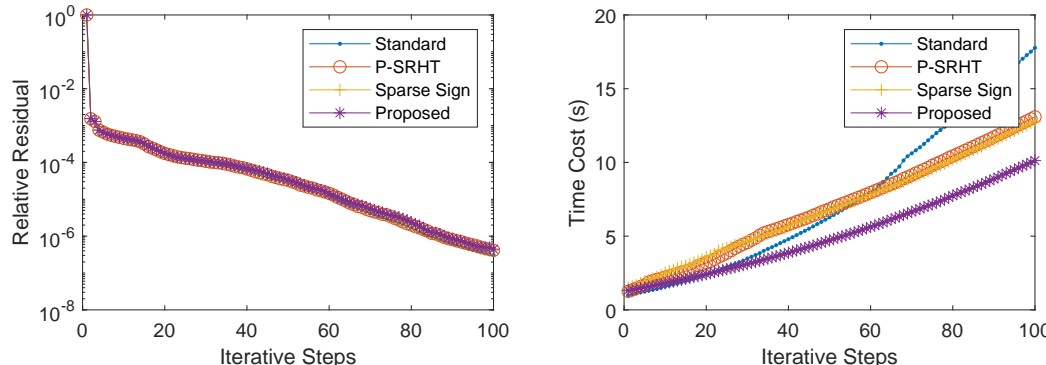

Figure 3: Convergence behaviors and runtime trends of the original GMRES and the randomized GMRES algorithms utilizing the randomized Arnoldi process (Alg. 2) with different embedding approaches for "ibmpg4t".

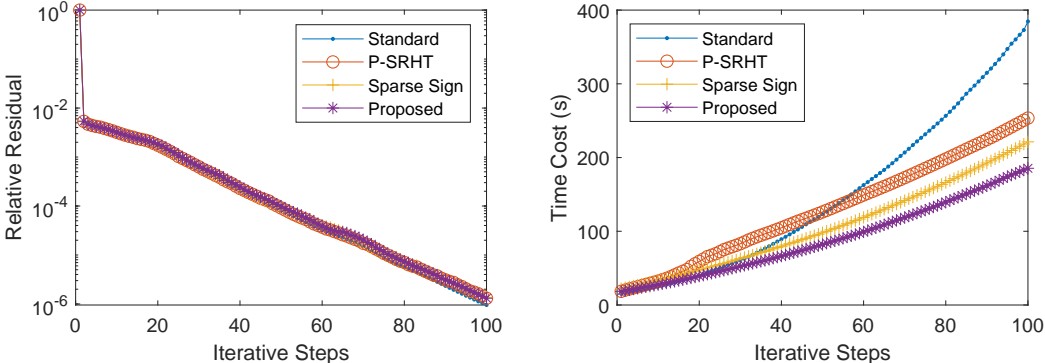

Figure 4: Convergence behaviors and runtime trends of the original GMRES and the randomized GMRES algorithms utilizing the randomized Arnoldi process (Alg. 2) with different embedding approaches for "thupg5".

The three parameters can be set independently. But different setting may lead to a slight difference in efficiency as well as accuracy. The runtime of execution stage of RLE is about $O(\xi\zeta n + \omega k^2)$. Therefore, if $n >> k^2$, the increase of $\xi$ and $\zeta$ can have a large impact on the runtime of RLE. Fig. 5 shows the experimental results on randomized GMRES leveraging the randomized Arnoldi process based on the RLE. From it we can see that, although different settings of parameters may have impact on the efficiency, the difference of accuracy caused by the varied parameters is negligible. With smaller $\xi$ and $\zeta$, i.e. ($\xi = 2$, $\zeta = 1$) or ($\xi = 1$, $\zeta = 2$), the runtime of GMRES is significantly reduced. The reason is that for this case $n = 4.7\text{E}6$ which is much larger than $k^2$ (i.e. $200^2$). Therefore, the increase of $\xi$ and $\zeta$ causes the increase of runtime of RLE.

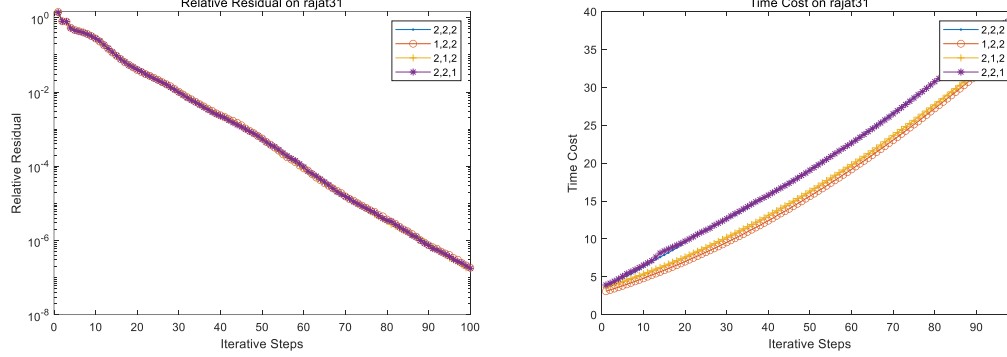

Figure 5: Performance of different settings of the three parameters (three numbers denotes $\xi$, $\zeta$, $\omega$, respectively. i.e "1,2,2" denotes $\xi = 1$, $\zeta = 2$, $\omega = 2$) on "rajat31".