# OpenReview forum: "A Rademacher-Like Random Embedding with Linear Complexity"
_ICLR.cc/2025/Conference — ICLR 2025 Conference Withdrawn Submission_

### Official Review · Reviewer_86Dx · 2024-10-27

**Soundness:** 2
**Presentation:** 1
**Contribution:** 1
**Rating:** 1
**Confidence:** 4

**Summary:**

This work considers the construction of structured random projections which can be applied very quickly to a vector for the purpose of solving least squares linear regression and quickly computing Arnoldi iterations for SVD. The construction of the random projection involves some combination of hashing into buckets and random signs, which are classic ideas in this literature. Experiments show that the proposed method offers faster application and lower approximation error than Gaussian embeddings and sparse embeddings.

**Strengths:**

The experimental results suggest that the authors’ methodology may have some promising applications in practice.

**Weaknesses:**

The theoretical analysis of the proposed random embedding only analyzes the expected squared norm of the embedding, and provides no insight into how the dimension $k$ of the embedding should scale with respect to the desired error level, success probability, or the space of vectors which are being preserved. Usually, these random embeddings have guarantees of the form “if the dimension $k$ is at least […], then the random map $S$ preserves the $\ell_2$ norm of all vectors $x$ in a set $X$ up to a factor of $(1+\epsilon)$ with probability at least $1-\delta$“ (see, e.g., Woodruff 2014).

Furthermore, this paper is missing important comparisons to prior work that is highly related to the ideas described in this paper. For example, CountSketch is a related construction which uses hashing and random signs that have applications to quickly solving low rank approximation and linear regression https://arxiv.org/abs/1207.6365.

The presentation is also quite poor, and I could not follow the precise definition of the proposed random embedding in Section 3.2.

**Questions:**

- Why do you specifically consider Arnoldi iterations for sketching SVD? There are many other methods for using sketching ideas in the context of low rank approximation: see, e.g., https://arxiv.org/abs/1207.6365 and https://arxiv.org/abs/1504.05477
- How do you choose the value of $k$ in your experiments? Your theorems don’t provide insight into how this parameter should be chosen.

---

> ### Author Response · Authors · 2024-11-25
>
> >**Weakness 1**: *The theoretical analysis of the proposed random embedding only analyzes the expected squared norm of the embedding, and provides no insight into how the dimension $k$ of the embedding should scale with respect to the desired error level, success probability, or the space of vectors which are being preserved. Usually, these random embeddings have guarantees of the form “if the dimension is at least […], then the random map $S$ preserves the $l2$ norm of all vectors $x$ in a set  $X$ up to a factor of $1+\epsilon$ with probability at least $1-\delta$“ (see, e.g., Woodruff 2014)*
>
> **Re**: Thank you for your comment. We present the analysis for the bound of oblivious subspace embedding in the revised paper (see Theorem 6). However, we would kindly point out that although theoretical analysis of the condition of oblivious subspace embedding is significant, we usually do not follow the theoretical bound to set $k$ or set up the embedding matrices.  This is similar to the application of sparse sign embedding. With the insight theoretical analysis in "Nearly tight oblivious subspace embeddings by trace inequalities (Cohen 2016)", a sparse sign matrix is an oblivious subspace embedding with constant distortion for an arbitrary $d$-dimensional subspace of $\mathbb{R}^{n}$ when $C=O(\log d)$ and $k=O(d\log d)$. However, in practice, the bound is still so large that one usually sets $k=\min(d,8)$ (for a detailed analysis please see Martisson 2020, Randomized numerical linear algebra: Foundations and algorithms). While the theoretical analysis is important for the performance under extreme conditions, in practice these conditions usually do not exist and we can just set the parameters to smaller values to achieve good performance.
>
> >**Weakness 2**: *Furthermore, this paper is missing important comparisons to prior work that is highly related to the ideas described in this paper. For example, CountSketch is a related construction which uses hashing and random signs that have applications to quickly solving low rank approximation and linear regression https://arxiv.org/abs/1207.6365.*
>
> **Re**: Thank you for your comment. We would kindly point out that Count Sketch is a special sparse sign embedding. For small $k$, Count Sketch may cause disastrous performance (see https://doi.org/10.1137/18M1201068, Tropp 2019). Tropp et al. have modified Count Sketch and proposed a practical sparse sign embedding in https://doi.org/10.1137/18M1201068. And we have compared our works with the Tropp's practical sparse sign embedding (https://doi.org/10.1137/18M1201068, Tropp 2019) in  experiments. Besides, we also reviewed the CountSketch algorithm in Section 1 and Section 2.1 in the revised paper.
>
> >**Weakness 3**: *The presentation is also quite poor, and I could not follow the precise definition of the proposed random embedding in Section 3.2.*
>
> **Re**: Thanks for your comment. Section 3 has been rewritten. We believe that the poor writing of the original paper causes misunderstandings and confusions. We apologize for that. We really hope that you can read the revised paper. Thanks again.
>
> >**Question 1**: *Why do you specifically consider Arnoldi iterations for sketching SVD? There are many other methods for using sketching ideas in the context of low rank approximation: see, e.g., https://arxiv.org/abs/1207.6365 and https://arxiv.org/abs/1504.05477*
>
> **Re**: Thanks for your comment. The two parts of experiments are independent and just show two example applications of our proposed algorithm. We are aware that there are other methods that uses sketching ideas in the context of low rank approximation, e.g. those you mentioned. I have cited them in the revised paper.
> The collaboration of the proposed method with them will be considered in our future work.
>
> >**Question 2**: *How do you choose the value of $k$ in your experiments? Your theorems don’t provide insight into how this parameter should be chosen.*
>
> **Re**: Thanks for your comment. In many practical applications, the bound of oblivious subspace embedding can be very large and is not applicable. Therefore, the works for practical applications often set $k$ first and calculate the error later to measure the effectiveness of proposed methods, instead of using the bound to calculate how $k$ should be set.  In the experiment of single-pass RSVD of https://doi.org/10.1137/18M1201068 (Tropp 2019), the $k$ is set to 10 and we follow this setting in our first experiment. For randomized Arnoldi process (Balabanov 2022), the authors do not recommend a specified setting of $k$. As the maximum number of Arnoldi steps is fixed to 100 in our experiment, we set $k=200$ (i.e. $k=2\times100$) to achieve good performance. In the revised paper, we have also added the analysis for the bound of oblivious subspace embedding (see Theorem 6), whose proof is given in Appendix A.2.

---

> > ### Comment · Reviewer_86Dx · 2024-11-26
> >
> > Thanks for the updated draft. I think I understand the claims better now. From the proof of the subspace embedding provided in Theorem 6 and Appendix A.2.2, I see that the authors intend for the embedding matrix to be a fully random dense (scaled) Rademacher matrix. However, there is then no structure to exploit between the rows, so the running time cannot be improved beyond O(nk). Are the other reviewers confident that the authors have indeed shown that one can multiply a dense k x n fully independent Rademacher embedding matrix and an n-dimensional vector in time O(n + k^2)?

---

> ### Author Response · Authors · 2024-11-27
> **Reply to Reviewer 86Dx**
>
> Notice that the proposed RLE approach does not keep the mutual independence of the matrix's entries (meaning any two disjoint subsets of entries are independent of each other). So, it is not a Rademacher embedding, and rather a Rademacher-like one.
>
> And,  in the proposed approach there is indeed structure/mechanism to share between the rows of Rademacher elements. That's the reason why the complexity of O(nk) is reduced. This is explained in Fig. 1 and the text of Section 3.2 (also Alg. 3).
>
> The time/space complexity of O(n + k^2) is rigorously derived as stated by Theorem 1 and its proof. Please read the part, if you have question.
> Thanks!

---

> ### Comment · Reviewer_86Dx · 2024-11-27
>
> What you are saying seems to be at odds with the claim in Theorem 3, which says that any two entries are independent. The norm preservation guarantees discussed in Theorem 6 also use this fact crucially. How do we reconcile this?
>
> I find Theorem 1 to be stated in a highly confounding way. Perhaps it would help if you can define the embedding matrix itself since this is the main part I don't understand, rather than defining it by how you execute its application to a vector. Alternatively, you could give a simple explanation of how the map acts on a vector, without immediately jumping into the optimized implementation. This would improve the readability.
>
> It seems the other reviewers are giving very high scores for this work, I wonder if any of the other reviewers have fully understood the construction and can help me understand?

---

> > ### Comment · Reviewer_PaoK · 2024-11-28
> >
> > Yes, I share the same confusion. On the one hand, the authors claim that  the elements of the proposed Rademacher-**Like** are  not independent; but on the other hand, they make the assumption of independence in subsequent analyses. It's a bit odd.

---

> > > ### Author Response · Authors · 2024-11-28
> > > **Reply to Reviewer 86Dx and Reviewer PaoK**
> > >
> > > Independence is a fundamental notion in probability theory. Two notions of independence need to be distinguished: **pairwise independence**, and **mutual independence**. The latter means **any two disjoint subsets of entries are independent of each other**. If you're not familiar to them, you can learn the concept from the wiki: https://en.wikipedia.org/wiki/Independence_(probability_theory).
> > >
> > >  In our Theorem 3, we say that any two entries are independent. This means **pairwise independence**. And, Theorem 3 also states arguments on mutual independence, but they are for the rows of $\Theta$, or for the entries in same row.
> > >
> > > For the whole matrix entries of $\Theta$, the mutual independence does not hold. This is the difference from the naive Rademacher matrix.

---

> > > > ### Comment · Reviewer_qHmq · 2024-11-28
> > > > **Refer to my review comments to the Authors**
> > > >
> > > > Dear Reviewers 86Dx and Paok,
> > > >
> > > > Weakness 3 in my Official Review comment is related and partially answers your questions, so you might want to see the discussion there.

---

> > > > > ### Comment · Reviewer_86Dx · 2024-11-28
> > > > >
> > > > > I understand the difference between pairwise (or in general k-wise) independence and full independence, this is a standard technique in the literature of sparse random projections (see, e.g., Theorem 2.6 in http://www.cs.cmu.edu/afs/cs/user/dwoodruf/www/wNow3.pdf). Note that you can indeed get an expectation bound with just 2-independence, but this is a very weak guarantee and is not useful for solving least squares regression, low rank approximation, etc. As in the previous reference, you would want at least 4-independence in the random signs to get even just a variance bound.
> > > > >
> > > > > This is the reason why I asked for more useful guarantees such as a subspace embedding. The authors then claimed a proof that the embedding is a subspace embedding in Appendix A.2.2. However, this proof uses full independence of the rows in line 708. Thus, my concerns remain unaddressed.

---

> ### Comment · Reviewer_qHmq · 2024-11-28
> **Good point, and but just typos?**
>
> Interesting and crucial point. The Authors should not have assumed mutual independence in line 708, the proof of Theorem 6, Appendix A.2.2. However, in the proof of Theorem 6, the Authors only consider the two-random-variable cross terms, so 2-pairwise independence would suffice if we just want to justify the proof?
>
> I do not fully understand the discussion in Theorem 2.6 in the paper you referred to, but they seem to evaluate four-random-variable cross terms in the proof. I guess what they want to evaluate there is essentially different from what the Authors have evaluated. Maybe, you (Reviewer 86Dx) want to claim that Theorem 2.6-type evaluation is important in the context of the Authors' research, though. At least, we could say the lack of such types of bounds should be discussed in the limitation section of the Authors' paper.

---

> > ### Comment · Reviewer_86Dx · 2024-11-28
> >
> > I politely disagree. The proof overview of Theorem 6 (a standard argument) is as follows: each row of the embedding $\Theta y$ is such that $\mathbb E(\Theta y)_i^2 = \lVert y\rVert_2^2$. If we then have roughly $d$ such rows, then by Chernoff bounds, the average of these $d$ rows gives an estimate of $\lVert y\rVert_2^2$ which is approximately accurate with probability $1 - \exp(-d)$. This allows us to union bound over a discretization of the unit L2 ball of size $\exp(d)$. Note that in the authors' proof, $(\Theta y)_i$ is $Q_i(\alpha)$ in equation (17) and some version of the Chernoff bounds is performed in Lemmas 1 and 2. To make such an argument go through, we need independence of theta $(\Theta y)_i$'s, which requires much more than just pairwise independence of the entries of $\Theta$.

---

> > > ### Comment · Reviewer_qHmq · 2024-11-28
> > > **Agree!**
> > >
> > > Now I get your point! (sorry, I originally only saw around line 708). Such Chernov-based bounds need converting the expectation of a product to the product of expectations, so correcting the proof to adapt to pairwise independent cases would be non-trivial, even if possible. I appreciate your discussion!

---

### Official Review · Reviewer_PaoK · 2024-10-29

**Soundness:** 2
**Presentation:** 2
**Contribution:** 3
**Rating:** 5
**Confidence:** 5

**Summary:**

The paper points out an interesting property of the Rademacher matrix: each pair of its rows has half of the column coordinates sharing the same sign, and the other half sharing different signs. This characteristic enables random embedding to efficiently compute only one row, with the result for the other row being readily inferable from the sign information.  To explicitly  determine the column coordinates sharing the same (and different) signs, the paper introduces a special matrix generation method, which allows random embedding implemented in linear complexity.

**Strengths:**

Compared to the original Rademacher matrix-based embedding, the proposed embedding method reduces computational costs, without accuracy loss on the RSVD and the randomized Arnoldi process.

**Weaknesses:**

1) The proposed matrix generation method is not easy to understand, as the presentation in section 3.2 lacks clarity and precision. Several crucial terms and symbols, such as "inner bucket," $w$, $ζ$, and $ k’$, remain undefined or unexplained. This section requires substantial improvement prior to publication.

2) The complexity (and stability) of the proposed matrix should heavily depend on several key parameters outlined in Algorithm 3, including $ξ$, $ζ$, $ω$, and $k’$. Regrettably, this relationship has been neglected, resulting in a theoretical shortcoming in the paper.

**Questions:**

1)  In the theoretical aspect, the authors focus on deriving the property of $E(\||y\||^2)=E(\||x\||^2)$  in Theorem 2. However, within the field of random embedding/projection, this property is already recognized as being applicable to random matrices with zero mean and unit variance. For details, see the reference:  Dimitris Achlioptas. 2001. Database-friendly random projections. In Proceedings of the twentieth ACM SIGMOD-SIGACT-SIGART symposium on Principles of database systems (PODS '01).

2) Line 396: The authors emphasize that the generated matrix is Rademacher-**like**. What distinguishes it from a traditional Rademacher matrix in mathematics? Both matrices take the values of $\pm 1$ with equal probabilities, appearing identical in this respect.

3) By the proposed method, how many rows can we potentially derive their values after computing just *one* row？ I think  this should be closely tied  to the parameters $ξ$, $ζ$, $ω$, and $k’$.  It is crucial to elucidate their relationships both  theoretically and practically.

---

> ### Author Response · Authors · 2024-11-25
>
> >**Weakness 1**: *The proposed matrix generation method is not easy to understand, as the presentation in Section 3.2 lacks clarity and precision. Several crucial terms and symbols, such as "inner bucket," $\omega$, $\zeta$, and $ k’$, remain undefined or unexplained. This Section requires substantial improvement prior to publication.*
>
> **Re**: Thanks for your comment. We have rewritten Section 3.2 and drawn a new Fig. 1 illustrating how our technique works. We really hope that you can review our modification. Thank you very much.
>
> >**Weakness 2**: *The complexity (and stability) of the proposed matrix should heavily depend on several key parameters outlined in Algorithm 3, including $\xi$, $\zeta$, $\omega$, and $k’$. Regrettably, this relationship has been neglected, resulting in a theoretical shortcoming in the paper.*
>
> **Re**: Thanks for your comment. we set $\omega$ as the multiple of $k'$ of $k$, which means that $k'=\omega k$. Therefore, there are just three *independent* parameters: $\xi$, $\zeta$ and $\omega$.
> We believe that it is enough for practice where these three paramters are independently set to very small values like 1 to 3. In our experiments, we just set  $\xi=2$, $\zeta=1$ and $\omega=2$ and obtain a very good result. From a qualitative point of view, larger values can offer better guarantee for stability while smaller values can offer better guarantee for speed. But just like the modified sparse sign matrices in https://doi.org/10.1137/18M1201068 (Tropp 2019), we believe that setting $\xi=2$, $\zeta=1$ and $\omega=2$ is good enough for some practical applications.
>
> >**Question 1**: *In the theoretical aspect, the authors focus on deriving the property of $E(||y||^2)=E(||x||^2)$ in Theorem 2. However, within the field of random embedding/projection, this property is already recognized as being applicable to random matrices with zero mean and unit variance. For details, see the reference: Dimitris Achlioptas. 2001. Database-friendly random projections. In Proceedings of the twentieth ACM SIGMOD-SIGACT-SIGART symposium on Principles of database systems (PODS '01).*
>
> **Re**: Thank you for your comment. Now we present more theoretical analysis of RLE. We keep the proof of $E(||y||^2)=E(||x||^2)$ only in the appendix now, because for RLE, the entries are only pairwise independent and thus we cannot directly leverage the conclustion in PODS'01. I have cited it in the revised paper, the proof of the theorem is moved to Appendix A.1.
>
> >**Question 2**: *Line 396: The authors emphasize that the generated matrix is Rademacher-like. What distinguishes it from a traditional Rademacher matrix in mathematics? Both matrices take the values of $\pm 1$ with equal probabilities, appearing identical in this respect.*
>
> **Re**:The biggest difference we believe is that we can not prove the mutual independence of the matrix's entries. Please read Section 3.3 of the revised paper.
>
> >**Question 3**: *By the proposed method, how many rows can we potentially derive their values after computing just one row? I think this should be closely tied to the parameters $\xi$, $\zeta$, $\omega$, and $k’$. It is crucial to elucidate their relationships both theoretically and practically.*
>
> **Re**: Sorry that the poor writing of our original paper causes the confusions. Please read Section 3.2 of the revised paper, and we think these confusions can be removed.

---

> > ### Comment · Reviewer_86Dx · 2024-11-27
> >
> > > Line 396: The authors emphasize that the generated matrix is Rademacher-like. What distinguishes it from a traditional Rademacher matrix in mathematics? Both matrices take the values of with equal probabilities, appearing identical in this respect.
> >
> > It seems like this reviewer shares my concern that it is unclear how the construction is different from a fully independent Rademacher matrix, in which case it should be impossible to apply the embedding to a vector in $O(n + k^2)$ time, barring a major breakthrough. Could this reviewer weigh in on this issue as well?

---

> ### Comment · Reviewer_PaoK · 2024-11-28
> **Response to the authors:  There already exist random matrices with lower complexity and easier implementation.**
>
> Thank you for the authors' efforts in revising the paper. The matrix construction method is now clearer. Unfortunately, it seems that the authors may not be familiar with the field of random projection. The paper's primary claim is that it reduces the complexity of random projection to $O(n)$. **However, there already exist random matrices capable of achieving a  complexity of $O(k\sqrt{n})$, which is smaller than $O(n)$ when $k<\sqrt{n}$.** Specifically, Li et al. have introduced a type of (0, +1, -1) random matrix, which has nonzero entries occuring with a probability of  only $1/\sqrt{n}$. Furthermore, such matrices are notably easier to implement in comparison to the method proposed by the authors.
>
> References: P. Li, T. J. Hastie, and K. W. Church. "Very sparse random projections." In KDD, 287-296, 2006.
>
> For future publications, the authors may consider identifying a specific application scenario to highlight the value of their proposed method.

---

> ### Author Response · Authors · 2024-11-28
> **Reply to Reviewer PaoK**
>
> We sincerely appreciate your new comments.
>
> The very sparse random projections that you mentioned is a type of sparse embedding similar to sparse sign embedding in, e.g.,
>
> PG Martinsson, JA Tropp, "Randomized numerical linear algebra: Foundations and algorithms", Acta Numerica, 2020.
>
> Its difference to the sparse sign embedding (or Count Sketch) is that the number of 1/-1 entries in the matrix's each row/column is stochastic (each entry has a probability of $1-1/\sqrt{n}$ to be 0). Therefore, it is not applicable for small $k$, because in that case there is a probability that a column of $\Theta$ is a zero vector making the the sketching result of sparse vector to be a zero vector, i.e. a disastrous result.
>
> Generally speaking, the very sparse random projection is less robust than the sparse sign embedding. Even though, the sparse sign embedding has the disadvantage for handling sparse vectors which may make the projected vectors lose linear independence (due to the sparsity of result), and the runtime overhead of irregular cache visiting due to sparse data structure.
>
> In contrast, the proposed Rademacher-like embedding inherits the advantages of dense embedding, like robustness, and still achieves a low complexity of $O(n+k^2)$.
>
> We will add this discussion in the revised paper.

---

### Official Review · Reviewer_aqrB · 2024-10-29

**Soundness:** 2
**Presentation:** 1
**Contribution:** 3
**Rating:** 5
**Confidence:** 5

**Summary:**

To reduce the complexity of the traditional Rademacher random embedding, this paper proposes a fast and stable Rademacher-like random embedding (RLE) that reduces the complexity from $O(kn)$ to $O(n)$ in time and space. Numerical experiments show that RLE enhances the efficiency and accuracy of single-pass randomized singular value decomposition (RSVD) and randomized Arnoldi process.

**Strengths:**

This study introduces a novel and interesting approach to reduce the complexity of  Rademacher  embedding, by exploring the inherent correlations between the rows of random Rademacher  matrices.

**Weaknesses:**

1)  The proposed method for matrix generation  is interesting, yet challenging to grasp due to its current presentation lacking of of vital details. As I see it, the authors intend to identify column indexes that share the same and distinct signs across **two** (or **more**?) rows during matrix generation. If the focus is solely on **two** rows, the proposed method  is **not** a necessity. Instead, we can simply generate a Rademacher matrix, and subsequently divide  its rows to  pairs. The column sign differences between each pair of rows can  then be easily identified.




2) Throughout the paper, the authors consistently stress the importance of the **stability** of random embedding matrices, which should be reflected in  the variance $Var(\||y|\|)$ given that  $E(\||y\||)=E(\||x\||)$. Nevertheless,  the variance  $Var(\||y|\|)$ is not explored in the paper. The authors restrict their attention to proving   $E(\||y\||)=E(\||x\||)$, a property already known  to hold for the random matrices with elements distributed symmetrically about zero.

**Questions:**

1) The method described in Section 3.2 is unclear and ambiguous, as it lacks detailed descriptions for the crucial parameters $\xi$, $\zeta$,  $k'$ and  $\omega$. For easier understanding, the introduction of the method  should be accompanied with the example illustrated in Figure 1. Furthermore, the example  needs to be modified to explicitly depict the relationships among  $\xi$, $\zeta$,  $k'$ and  $\omega$.

2) It is unclear how to determine the optimal values of $\xi$, $\zeta$,  $k'$ and $\omega$. To tackle this issue, it is suggested to investigate  the effects of these parameters on the complexity and stability of the Rademacher-Like random embedding.

3) The proposed matrix achieves linear complexity $O(n)$  when $k<\sqrt{n}$, which is equivalent to a compression ratio $k/n<1/\sqrt{n}$.  This condition implies that the  method is only applicable in cases of very low compression ratios, i.e. $k/n=1/100$ for $n=10000$. However, random embedding typically performs poorly in these scenarios due to significant information loss and instability. In summary, the condition  $k<\sqrt{n}$ may restrict the practical application of the proposed method.

4) It appears that the proposed method does not support parallel computation, given its row-by-row operation mechanism.




There are many minor errors in the paper, and here is a small selection of them.

Line 112: the dimensions of $x$ and $\theta$

Line 158: the sketching process of ?

Line 230: two row of $y$

Line 250: the term “inner buckets” emerges without definition.

Line 254: in each outer buckets



Line: 262: each entries



Line 633: theoretical result

Line 779: outer 2 buckets

---

> ### Author Response · Authors · 2024-11-25
>
> >**Weakness 1**: *The proposed method for matrix generation is interesting, yet challenging to grasp due to its current presentation lacking of of vital details. As I see it, the authors intend to identify column indexes that share the same and distinct signs across two more? rows during matrix generation. If the focus is solely on two rows, the proposed method is not a necessity. Instead, we can simply generate a Rademacher matrix, and subsequently divide its rows to pairs. The column sign differences between each pair of rows can then be easily identified.*
>
> **Re**: Thank you for your comment. We have rewritten Section 3 (especially Section 3.2) and we believe the modified version is much easier to understand. The new Fig. 1 could better illustrate the mechanism of our proposed RLE embedding.
>
>
> >**Weakness 2**: *Throughout the paper, the authors consistently stress the importance of the stability of random embedding matrices, which should be reflected in the variance $Var(||y||)$ given that $E(||y||)=E(||x||)$. Nevertheless, the variance is not explored in the paper. The authors restrict their attention to proving  $E(||y||)=E(||x||)$, a property already known to hold for the random matrices with elements distributed symmetrically about zero.*
>
> **Re**: Thank you for your comment. Now we present more theoretical analysis on RLE, including the bound for oblivious subspace embedding (Theorem 6).
> And, the term stability in the original paper has more meaning on robustness. So, we change it to the latter in the revise paper.
> Besides, we keep the proof of  $E(||y||^{2})=E(||x||^{2})$ only in the appendix.
>
> >**Question 1**: *The method described in Section 3.2 is unclear and ambiguous, as it lacks detailed descriptions for the crucial parameters $\xi$, $\zeta$, $k'$ and $\omega$. For easier understanding, the introduction of the method should be accompanied with the example illustrated in Figure 1. Furthermore, the example needs to be modified to explicitly depict the relationships among $\xi$, $\zeta$, $k'$ and $\omega$.*
>
> **Re**:  Thanks for your comment. In the revised paper, the method should have been well explained with Fig. 1 and Alg. 1. We have completely rewritten Section 3.2.
> Besides, there are actually three parameter, as $k'=\omega k$. Their meaning and value range are also explained in Section 3.2.
>
> >**Question 2**: *It is unclear how to determine the optimal values of $\xi$, $\zeta$, $k'$ and $\omega$. To tackle this issue, it is suggested to investigate the effects of these parameters on the complexity and stability of the Rademacher-Like random embedding.*
>
> **Re**: Thank you for your comment. We have done an ablation study on the parameters in Appendix A.4. For the relationship of those three parameters $\xi$, $\zeta$ and $\omega$, they can be set independently. Moreover, each of them can be set to a small value like from 1 to 3, which produces
> good enough results. In our experiments, we just set  $\xi=2$, $\zeta=1$ and $\omega=2$. Generally speaking, larger values can offer better guarantee for stability while smaller values can offer better guarantee for speed.
>
> >**Question 3**: *The proposed matrix achieves linear complexity $O(n)$ when $k<\sqrt{n}$, which is equivalent to a compression ratio $\frac{k}{n}<\sqrt{n}$. This condition implies that the method is only applicable in cases of very low compression ratios, i.e. $\frac{k}{n}=\frac{1}{100}$ for $n=10000$. However, random embedding typically performs poorly in these scenarios due to significant information loss and instability. In summary, the condition $k<\sqrt{n}$ may restrict the practical application of the proposed method.*
>
> **Re**: Thank you for your comment. Actually, as long as $k$ is not larger than $O(\sqrt{n})$, the linear complexity still holds. Even if $k$ is larger, the proposed method still can have less computational cost than the Rademarcher embedding because $k<n$.
>
> >**Question 4**: *It appears that the proposed method does not support parallel computation, given its row-by-row operation mechanism.*
>
> **Re**: Thank you for your comment. We have made a discussion on this in the end of our main text. The proposed RLE approach has the potential to be well parallelized.
>
> >**Question 5**: *There are many minor errors in the paper, and here is a small selection of them.*
>
> **Re**: Thank you for your comment. We have corrected them in the revised paper.

---

> ### Comment · Reviewer_aqrB · 2024-11-28
>
> Thank you for your response.  I still need clarification on several issues.
>
> Regarding Weakness 2: The analysis of  the "variance" is necessary in the study of random projection, as it indicates how well the projected data preserves the pairwise distance of the original data. This distance preservation capability is vital in determining the efficacy of random projection in practical applications, such as data classification.
>
> Regarding Question 3:  What I intended to convey is that the compression ratio $\frac{k}{n}<\sqrt{n}$ required in the analysis is too low, which suggests a  weak distance preservation property and  consequently a significant data information loss.  In this case, it is hard to achieve a satisfactory/meaningful performance in  downstream applications, like data classification. This undermines the practical application value of the proposed method.
>
> Regarding matrix generation:  Compared to traditional random projection, the proposed method exhibits higher implementation complexity, achieving a reduction in computational complexity by sacrificing control complexity. However, whether this trade-off results  really contribute to reducing computational resources or time remains a matter that requires further in-depth analysis or proof.

---

### Official Review · Reviewer_qHmq · 2024-11-02

**Soundness:** 4
**Presentation:** 2
**Contribution:** 4
**Rating:** 3
**Confidence:** 2

**Summary:**

The authors have proposed a novel random projection algorithm from a $n$-dimensional vector to $k$-dimensional vector, where the implicitly implemented random projection matrix is Rademacher-like in the sense that each entry is in $\\{-1, +1\\}$ and its computational complexity is $O(n+k^2)$, not $O(nk)$.

**Strengths:**

1. The explanation of the background and motivation of the area is clear. It is designed so that even readers not in the random embedding can understand the motivation.
2. The novelty and significance of the proposed method are also clearly presented.
3. Non-triviality is not necessary for machine learning papers in general. Having said that, it is notable that one would not easily come up with the essence of the trick used in this paper to avoid explicit matrix multiplication operations. Hence, this paper has the potential to be an essential building block of future innovative papers.
4. The base idea of avoiding explicit random matrix generation to avoid $O(nk)$ complexity is intuitively explained in Section 3.1. This helps readers understand the non-trivial phenomenon achieved by the proposed algorithm.
5. The numerical experiments have successfully shown that what the proposed algorithm sacrificed is not practically problematic.

**Weaknesses:**

Overall, most parts of the paper are excellent, but the core explanation of the algorithm could be refined, which is the main reason for my current low score.
1. Section 3.2. did not make sense to me. Possible reasons are that words like "inner buckets" or "outer buckets" do not help intuitive understanding. If there is no analogue between the algorithm and the daily usage of the buckets, it would rather be better to use specific mathematical symbols instead. Specifically, it would be better to explain each operation in Section 3.2. using symbols appearing in Algorithms 3 and 4, e.g., $R,C,P,E$ or entries in those matrices.
2. I could not understand Figure 1, either. A possible reason is that readers cannot understand from figures what each ball or curve connecting two balls means. I encourage the Authors to explain the algorithm using specific numerical examples instead of Figure 1.
3. The current manuscript does not clarify what the proposed algorithm sacrifices. If I understand it correctly, the entries in an implicit project matrix generated by the proposed algorithm are pairwisely independent but not mutually independent, while the entries in a naive Rademacher projection matrix are always mutually independent. Clarifying this sacrifice (perhaps as a limitation) must benefit readers. At least it helps readers understand why the proposed embedding is called "Rademacher-**like**," not "Rademacher." If the proposed algorithm did not sacrifice anything theoretically from naive Rademacher random projection, numerical experiments would not be needed.
4. The Authors emphasise that the proposed algorithm is "linear". This is not ideal, since naive Rademacher random embedding is also linear with respect to each of $n$ or $k$. My brain cannot come up with a better natural language word, but you might want to emphasize the mathematical difference between $O(nk)$ and $O(n+k^2)$. At least, the expression of $O(n+k^2)$ should appear somewhere.
5. Most parts of the mathematical explanation in the paper are clear, but the lack of the definition of $d$ makes Section 2.1 hard to understand.

Typos:
- $x \\in \\mathbb{R}$ -> $x \\in \\mathbb{R}^n$ in line 112.

---
**After discussion period**:
In the discussion with Reviewer 86Dx, we have found that the current proof of Theorem 6 uses mutual independence while it is not the case in the proposed Rademacher-like representations. Regrettably, I have lowered the score.

**Questions:**

1. Could you clarify the definition of $d$?
2. Could you explain each step of the algorithm using specific numeric examples and associate each step with entries in $R, C, P, E$? The unclarity of this part is the main reason for the current low score.
3. Comment: Theorem 3 states that entries in $\\Theta$ are pairwisely independent. We should leave the current expression of Theorem 3 also since readers might not know the phrase "pairwise independent," but including the phrase "pairwise independent" would help you clarify what the proposed algorithm sacrifices compared to the naive Rademacher random embedding, whose projection matrix's entries are mutually independent, not only pairwise independent.

---

> ### Author Response · Authors · 2024-11-25
>
> >**Weakness 1**: *Overall, most parts of the paper are excellent, but the core explanation of the algorithm could be refined, which is the main reason for my current low score.  Section 3.2. did not make sense to me. Possible reasons are that words like "inner buckets" or "outer buckets" do not help intuitive understanding. If there is no analogue between the algorithm and the daily usage of the buckets, it would rather be better to use specific mathematical symbols instead. Specifically, it would be better to explain each operation in Section 3.2. using symbols appearing in Algorithms 3 and 4, e.g. $R,C,P,E$, or entries in those matrices.*
>
> **Re**: Thank you for your comment. We have rewritten Section 3.2. We believe that the modified version is much easier to understand.
>
> >**Weakness 2**: *I could not understand Figure 1, either. A possible reason is that readers cannot understand from figures what each ball or curve connecting two balls means. I encourage the Authors to explain the algorithm using specific numerical examples instead of Figure 1.*
>
> **Re**: Thank you for your comment. We have drawn a new figure in Section 3.2. We believe that the new figure will resolve the problem.
>
>
> >**Weakness 3**: *The current manuscript does not clarify what the proposed algorithm sacrifices. If I understand it correctly, the entries in an implicit project matrix generated by the proposed algorithm are pairwisely independent but not mutually independent, while the entries in a naive Rademacher projection matrix are always mutually independent. Clarifying this sacrifice (perhaps as a limitation) must benefit readers. At least it helps readers understand why the proposed embedding is called "Rademacher-like," not "Rademacher." If the proposed algorithm did not sacrifice anything theoretically from naive Rademacher random projection, numerical experiments would not be needed.*
>
> **Re**: Thank you for your comment. Indeed, we can not prove the mutual independence of our proposed embedding matrices although we have proven the pairwise independence of it. That's why we call it Rademacher-like embedding and we have written the detailed reason of it in the revised paper. Additionally, the theoretical analysis reveals the proposed RLE share the same oblivious subspace embedding bound with the Rademacher embedding (see Theorem 6).
>
> >**Weakness 4**: *The Authors emphasise that the proposed algorithm is "linear". This is not ideal, since naive Rademacher random embedding is also linear with respect to each of $n$ or $k$. My brain cannot come up with a better natural language word, but you might want to emphasize the mathematical difference between $O(nk)$ and $O(n+k^{2})$. At least, the expression of  should appear somewhere.*
>
> **Re**: Thank you for your comment. We have replaced the expression of linear complexity or $O(n)$ with $O(n+k^{2})$ in most places of the paper. And, we also explained that if $k$ is not larger than $O(\sqrt{n})$, it means the linear complexity.
>
> >**Weakness 5 and Question 1**: *Most parts of the mathematical explanation in the paper are clear, but the lack of the definition of $d$ makes Section 2.1 hard to understand. Could you clarify the definition of  $d$?*
>
> **Re**: Thank you for your comment. $d$ is the dimension of the subspace of $\mathbb{R}^{n}$. The part was not well written. In the revised paper, we have rewritten it.
>
> >**Question 2**: *Could you explain each step of the algorithm using specific numeric examples and associate each step with entries in $R,C,P,E$? The unclarity of this part is the main reason for the current low score.*
>
> **Re**: Thank you for your comment. We have rewritten Section 3.2 with a new Fig. 1, which serves as an example. Besides, E is now two tensors, i.e. E and S.
> If you have any further questions about the revised version, please do not hesitate to ask us.
>
> >**Question 3**: *Comment: Theorem 3 states that entries in $\Theta$
>  are pairwisely independent. We should leave the current expression of Theorem 3 also since readers might not know the phrase "pairwise independent," but including the phrase "pairwise independent" would help you clarify what the proposed algorithm sacrifices compared to the naive Rademacher random embedding, whose projection matrix's entries are mutually independent, not only pairwise independent.*
>
> **Re**: Thank you for your comment. Indeed, we can not prove the mutual independence of the embedding matrix's entries in the proposed RLE approach. That's why we call it Rademacher-like embedding.
> It should be pointed out, the theoretical analysis reveals it can achieve the same bound as Rademacher embedding (see the new Theorem 6 in the revised paper).

---

> > ### Comment · Reviewer_qHmq · 2024-11-26
> > **Thank you for the detailed responses but still some unclearity remains**
> >
> > I appreciate the Authors' detailed responses and revisions. I feel the revised version of Section 3.2. is much better than before. I will read your response carefully and change the score appropriately during the reviewer discussion phase if needed.
> >
> > Questions:
> > - Could you clarify the role of $C$, which is not explained even in the body text of Section 3.2. in the revised version?
> > - You said $\\zeta$ and $\\xi$ can be small in the revised version. Do we have formal conditions for theorems to hold? Are $\\zeta=1$ or $\\xi=1$ allowed?

---

> ### Author Response · Authors · 2024-11-27
> **Reply to  Reviewer qHmq**
>
> Thank you very much for your new comments!
>
> The role of $C$ is for randomly selecting a column in $sum$ for random splitting of the terms for calculating $Px$. I just added the explanation in the Section 3.2 of the paper. It reads,
>
> ``Matrix $C\in \mathbb{Z}^{\xi\times n}$ is for realizing the random splitting, i.e. the $j$-th term in the sum for calculating $Px$ is accumulated into the $C(l,j)$-th column of $sum(l,:,:)$.''
>
> We have carefully checked the six theorems. They hold for any positive numbers of $\zeta$ and $\xi$. Yes, $\zeta=1$ and $\xi=1$ are allowed. From the ablation study in Appendix A.4 we see that setting smaller $\zeta$ and $\xi$ brings runtime reduction for the application to randomized Arnoldi process, without sacrifice on accuracy. But generally speaking, larger $\zeta$ and $\xi$ should enhance more randomness and the robustness of the proposed RLE embedding.
>
> We also revised the paper, and updated it.

---

> > ### Comment · Reviewer_86Dx · 2024-11-27
> >
> > > The current manuscript does not clarify what the proposed algorithm sacrifices. If I understand it correctly, the entries in an implicit project matrix generated by the proposed algorithm are pairwisely independent but not mutually independent, while the entries in a naive Rademacher projection matrix are always mutually independent. Clarifying this sacrifice (perhaps as a limitation) must benefit readers. At least it helps readers understand why the proposed embedding is called "Rademacher-like," not "Rademacher." If the proposed algorithm did not sacrifice anything theoretically from naive Rademacher random projection, numerical experiments would not be needed.
> >
> > It seems like this reviewer shares my concern that it is unclear how the construction is different from a fully independent Rademacher matrix, in which case it should be impossible to apply the embedding to a vector in $O(n + k^2)$ time, barring a major breakthrough. Could this reviewer weigh in on this issue as well?

---

> > > ### Comment · Reviewer_qHmq · 2024-11-28
> > > **Perhaps, my review comments helps (pairwise vs mutual independence)**
> > >
> > > Thank you for starting the discussion. I do not see a very strange phenomenon here at the moment, and my review comment has already clarified, which the Authors agreed with. That is, the entries in the implicit projection matrix are **pairwise independent** but not **mutually independent.** This is a kind of sacrifice the authors made to achieve a complexity better than $O(nk)$. So, my understanding is that the proposed algorithm gave up mutual independence and only achieved pairwise independence, but mutual independence was not necessary for success in practical applications (at least for some cases) as long as pairwise independence and other properties are maintained.
> > >
> > > Hope it answers your questions, at least partially.

---

### Author Response · Authors · 2024-11-25

First of all, We sincerely thank all reviewers for their helpful comments. With the help of those comments, we have done **a major revision** on our manuscript and we believe our manuscript has been greatly improved. All modifications in the original manuscript are marked in ``BLUE``. The modifications can be summarized as follows.

1. We have replaced most expressions of linear complexity or $O(n)$ in the paper with $O(n+k^2)$ complexity, because the latter is a more rigorous presentation. We also emphasize that if $k$ is not large than $O(\sqrt{n})$, it is just $O(n)$ complexity.

2. We have rewritten Section 3.2 to better explain the proposed Rademarcher-like embedding (RLE) approach, with a new Figure 1 replacing the old one. Figure 1 is served as an example for explaining RLE. This explanation is closely based on the algorithm for the execution phase of RLE, and discards the vague expressions of buckets and random sequences. Due to the limit of space, Alg. 3 (for the setup phase) and Lemma 1 (which is less relevant) in the original paper are removed. The setup phase is straightforward, and is now explained by the second to last paragraph of Section 3.2.




3. Theoretical analysis on the proposed RLE is reorganized and largely modified in Section 3.2. Now there are six theorems, where Theorem 3 extends that in the original paper and Theorem 6 on the oblivious subspace embedding bound is totally new.


4. An ablation study on the three parameters ($\xi, \zeta, \omega$) is added with the results presented by Fig. 5 in Appendix A.4.
And, a discussion on how to parallelize RLE is presented at the end of the main text.


5. The Appendix is also reorganized, with irrelevant content removed and new contents like the proof of Theorem 6 and result of  ablation study added.


6. Finally, 4 references are added, and the whole manuscript is carefully proofread.

---

> ### Author Response · Authors · 2024-11-27
> **Revision of the paper**
>
> Just uploaded the paper with several minor revisions according to the reviewers' new comments, including adding a reference.
>
> The new revisions are colored in **red**.

---

### Note · Authors · 2025-02-14

**Comment:**

We would like to thank PC, SAC, AC, reviewers again for their valuable feedbacks.
Sorry for missing the revision deadline of paper, and the latest submitted paper truly needs a major revision.
Best Regards.

**Withdrawal Confirmation:**

I have read and agree with the venue's withdrawal policy on behalf of myself and my co-authors.

---

### Meta-Review · Area_Chair_ZeTr · 2024-12-05

**Metareview:**

This paper aims to develop theoretical guarantees for random projection where the projection matrix moves away from Rademacher. Authors proposed an approach that reduces the computational time while almost preserving approximation guarantees.

Reviewers agree that this work is interesting, and the generalization beyond Rademacher is important.

However, reviewers also found that the paper requires a major revision to improve the presentation, and that some theoretical results are questionable.

**Additional Comments On Reviewer Discussion:**

Reviewers raised concerns about clarity. During the discussion phase, authors addressed some of them.

Reviewer 86Dx pointed critical errors in the proof. This leads to a clear reject.

---

### Decision · Program_Chairs · 2025-01-22

Reject